METHODS AND RESOURCES

# A comparison of viral strategies and model systems to target norepinephrine neurons in the locus coeruleus reveals high variability in transgene expression patterns

Chantal Wissing[1,2☙], Lena S. Eschholz[1,3☙], Maxime Maheu[1,4], Kathrin Sauter[1], Fabio Morellini[5], J. Simon Wiegert[1,3*], Alexander Dieter[1,3*]

1 Synaptic Wiring Lab, Center for Molecular Neurobiology, University Medical Center Hamburg-Eppendorf, Hamburg, Germany, 2 Institute for Systems Physiology, Faculty of Medicine, University Clinic, Cologne, Germany, 3 Department of Neurophysiology, MCTN, Medical Faculty Mannheim, Heidelberg University, Mannheim, Germany, 4 Section Computational Cognitive Neuroscience, Department of Neurophysiology and Pathophysiology, University Medical Center Hamburg-Eppendorf, Hamburg, Germany, 5 Research Group Behavioral Biology, Center for Molecular Neurobiology, University Medical Center Hamburg-Eppendorf, Hamburg, Germany

☙ These authors contributed equally to this work.
* simon.wiegert@medma.uni-heidelberg.de (JSW); alexander.dieter@medma.uni-heidelberg.de (AD)

## Abstract

The locus coeruleus (LC) norepinephrine (NE) system is involved in a variety of physiological and pathophysiological processes. Refining our understanding of LC function largely relies on selective transgene expression in LC-NE neurons, allowing targeted manipulation and readout of noradrenergic neurons. Here, we performed a side-by-side comparison of the most commonly used strategies to genetically target the LC, including different cre driver lines and promoter-mediated transgene expression. We report differences between these strategies in terms of transgene expression efficacy and specificity. Parallelly, we found no behavioral alterations in cre-expressing mice of any mouse line compared to wild-type littermates. Finally, to further facilitate the investigation of LC-NE function, we created a suite of constructs, including a reporter protein, a calcium indicator, and a light-driven cation channel, whose expression is mediated by the previously described PRS×8 promoter. These constructs allow identification, monitoring, and manipulation of LC-NE activity either in wild-type mice, or in combination with tissue-specific manipulations of different cre driver lines. The results of our study are crucial for the interpretation of previous experiments using the respective targeting strategies, as well as for the design of future studies.

**Data availability statement:** Numerical data underlying all figures and supplementary figures can be found in S1 Data. Raw confocal data, Z-Stacks, and cell masks annotated by CellPose (presented in Figs 1C–1F, 3B–3D, S2, and S7) as well as raw and pre-processed epifluorescence data (presented in Figs 2 and 3F/3G/3I/3J) is available at https://gin.g-node.org/SW_lab/Wissing_and_Eschholz_et_al_PLOS_Biology_2025 (https://doi.gin.g-node.org/10.12751/g-node.msn52q/). The code generated during this study is available at https://github.com/SynapticWiringLab/Wissing_and_Eschholz_LC-transduction/ (https://doi.org/10.5281/zenodo.15474852). The custom-trained CellPose model used to segment LC neurons in this study (CP4LC_Wissing_Eschholz_et_al) can be found in S1 Material, as well as on the g-node repository mentioned above (https://doi.gin.g-node.org/10.12751/g-node.msn52q/).

**Funding:** This work was supported by the Deutsche Forschungsgemeinschaft (SFB 936, grant agreement: 178316478; project B8 to JSW and B7 to FM; https://www.dfg.de/de). MM is supported by the Alexander von Humboldt Stiftung, and by the Fondation Bettencourt-Schueller. The funders had no role in study design, data collection and analysis, decision to publish, or preparation of the manuscript.

**Competing interests:** The authors have declared that no competing interests exist.

**Abbreviations:** BAC, bacterial artificial chromosome; CAV-2, canine Adenovirus type 2; DBH, dopamine-β-hydroxylase; DIO, double-floxed inverted open reading frames; eGFP, enhanced green fluorescent protein; EPM, elevated plus maze; hSyn, human synapsin; LC, locus coeruleus; MWM, Morris Water Maze; NE, norepinephrine; NET, norepinephrine transporter; NGS, normal goat serum; PBS, phosphate-buffered saline; PRS, Phox2a/Phox2b response site; TH, tyrosine hydroxylase; VTA, ventral tegmental area.

## Introduction

The locus coeruleus (LC)—a bilateral nucleus located in the brainstem—is the major source of norepinephrine (NE; also: noradrenaline) in the central nervous system. Despite its small size (~3,000 neurons in the rodent brain [1]), the LC is involved in several physiological functions, including alertness, arousal, sensory perception, attention, learning, and memory consolidation [2–4]. In addition, the LC has been implicated in numerous pathological conditions including depression, post-traumatic stress disorder, Alzheimer's disease, and Parkinson's disease [2–6]. Accordingly, the LC-NE system is currently the subject of detailed investigations [2,3,7–9]. Recent research suggests this nucleus is modular, with subpopulations of cells targeting different brain regions, presumably with different physiological roles [2,10,11]. Thus, the LC may exert a more sophisticated, target-specific modulation of brain activity than previously thought. These findings have brought the LC-NE system into the focus of many research groups that rely on the specific targeting of LC-NE neurons to interrogate LC function [2].

Key technologies to access the LC-NE system in animal models include genetic tools such as cre driver lines (S1 Table) and viral transduction, either with conditional constructs, or with tissue-specific promoters. These methods allow gene expression in molecularly identified neural populations, even in a projection-specific manner [12]. Detailed studies of the LC-NE system are more feasible than ever with the recent development of genetically encoded sensors for NE [13,14] and dopamine (DA) [15–17], as well as opto- or chemogenetic strategies that allow the manipulation of LC subpopulations in a projection-specific manner. Driver lines have been developed to express cre recombinase under the control of the dopamine-β-hydroxylase (DBH) promoter in mice [18,19] and rats [20], under control of the norepinephrine transporter (NET) in mice [21], and several mouse [18,22,23] and rat [24] driver lines exist that express cre recombinase under the control of the tyrosine hydroxylase (TH) promoter. DBH catalyzes the synthesis of NE from DA and is therefore specific for NE neurons. Similarly, NET, the membrane protein that mediates reuptake of extracellular NE, is used as a selective marker for NE-releasing neurons. In contrast, TH is the enzyme involved in the synthesis of L-DOPA from tyrosine, three reaction steps upstream of the end product NE, and is therefore present in all catecholaminergic cells, including dopaminergic ones.

Alternative approaches to target the LC-NE system that do not rely on transgenic lines have also been proposed. In particular, NE-specific transgene expression has been achieved using a minimal synthetic promoter, containing 8 copies of a *cis*-regulatory element (Phox2a/Phox2b response site; PRS) derived from the human *Dbh* promoter, thus termed PRS×8 [25]. This promoter has been shown to be specific for noradrenergic cells in both mice [26,27] and rats [25,28]. While the four strategies (*Dbh^cre^*, *Net^cre^*, *Th^cre^*, and PRS×8) for targeting the LC-NE system provide a wide range of alternatives to choose from, they come with different caveats and challenges (see, e.g., [29–31], for reviews on transgene expression strategies). Critically, a direct comparison of the most commonly used LC-NE targeting strategies is lacking, and proper quantification of viral transduction is not always reported.

Here, we performed a side-by-side comparison of virus-mediated transgene expression in the LC-NE system by introducing cre-dependent reporter genes into the LC of *Dbh^cre^*, *Net^cre^*, and *Th^cre^* mice, and unconditional reporter genes under the control of the PRS×8-promoter into the LC of wild-type mice. We report substantial differences in transgene expression, both in terms of efficacy and molecular specificity, while we did not detect any behavioral changes associated with cre expression in any of the mouse lines tested. To facilitate investigations of the LC-NE system, we have developed a suite of PRS×8-driven tools including a fluorophore (eGFP), a calcium indicator (jGCaMP8m), and a red-light sensitive channelrhodopsin (ChrimsonR) that allow tissue-specific transgene expression. These tools can be used either in wild-type mice or in combination with cre driver lines, thereby facilitating simultaneous transgene expression in additional neuronal populations [32]. Our findings have broad implications for researchers using genetic manipulation of the noradrenergic system to study LC function: they may guide the choice of an appropriate targeting strategy in future research, and may assist in the interpretation of data obtained from the respective model system.

## Results

### Efficacy and specificity of transgene expression across model systems

To visualize transgene expression in the LC-NE system, we bilaterally injected the LC with titer-matched suspensions of recombinant adeno-associated virus (rAAV2/9) encoding enhanced green fluorescent protein (eGFP; Fig 1A). In *Dbh^cre^*, *Net^cre^*, and *Th^cre^* mice, cre-dependent gene expression was controlled by combining double-floxed inverted open reading frames (DIOs) [33] of the transgene with a strong, synthetic promotor (CAG) [34]. In wild-type mice (C57BL/6J), the reporter gene was expressed under the control of the synthetic PRS×8 promoter. Six weeks after injection, we analyzed eGFP expression by fluorescence microscopy of coronal sections which were immuno-stained against TH (to visualize NE-expressing neurons) and GFP (to enhance the signal originating from transgene expression; Fig 1B). To quantify transgene expression at a cellular level, cells expressing TH (TH⁺) or eGFP (eGFP⁺) were automatically segmented using the deep learning-based algorithm *CellPose* [35,36] (Fig 1C).

To assess the ability to selectively target the LC, we evaluated the *efficacy* and *specificity* of viral transduction of LC-NE neurons. To this end, we compared the overlap between masks segmented in the TH and GFP channels, respectively, defining cells with an overlap of ≥50% as co-expressing. Efficacy is defined as the proportion of TH⁺ cells co-expressing eGFP, while specificity reports the proportion of eGFP⁺ cells co-expressing TH.

We found differences in the efficacy between model systems (one-way ANOVA, $F_{24} = 14.71$, $p = 1.2 \times 10^{-5}$; Fig 1D): while no significant differences in efficacy were detected between *Dbh^cre^*, *Net^cre^*, and PRS×8-mediated transgene expression ($70.5 \pm 11.8\%$, $79.5 \pm 9.0\%$, $78.2 \pm 12.9\%$; mean ± SD; Tukey's test, $p = 0.68/0.77/0.99$ for *Dbh^cre^* versus *Net^cre^*/*Dbh^cre^* versus PRS×8/*Net^cre^* versus PRS×8), the efficacy of *Th^cre^* mediated expression was significantly lower than the other model systems ($33.3 \pm 22.7\%$; Tukey's test, $p = 6 \times 10^{-4}/3 \times 10^{-5}/5 \times 10^{-5}$, compared against *Dbh^cre^*/*Net^cre^*/PRS×8, respectively). We further noted a large variability in the efficacy of eGFP expression in *Th^cre^* mice, with one mouse showing an efficacy of up to 75%, while the efficacies of other mice—even from the same litter—were below 20% (S1 Fig).

We also found differences in the specificity (one-way ANOVA, $F_{24} = 14.47$, $p = 1.4 \times 10^{-5}$) between model systems (Fig 1E): Here, the specificity of *Th^cre^* mediated expression ($46.0 \pm 12.1\%$) was significantly lower than *Dbh^cre^* ($82.2 \pm 9.5\%$), *Net^cre^* ($71.4 \pm 13.6\%$), and PRS×8 ($65.2 \pm 5.0\%$) mediated transgene expression (Tukey's test, $p = 7 \times 10^{-6}/ 7 \times 10^{-4}/0.01$ *Dbh^cre^*/*Net^cre^*/PRS×8 against *Th^cre^*). Furthermore, the *Dbh^cre^* approach was significantly more specific than PRS×8 mediated transgene expression (Tukey's test, $p = 0.03$). No significant differences were found between *Dbh^cre^* versus *Net^cre^* and between PRS×8 versus *Net^cre^*.

In addition to differences in the efficacy of transgene expression and its specificity to TH⁺ neurons, we realized that also the levels of transgene expression in relation to the levels of endogenous TH expression differed across approaches. While the averaged normalized eGFP fluorescence in true positive neurons (i.e., neurons that co-expressed eGFP and TH) was expectedly higher than in false positive neurons (i.e., neurons that expressed eGFP but not TH) in *Dbh^cre^*, *Net^cre^*

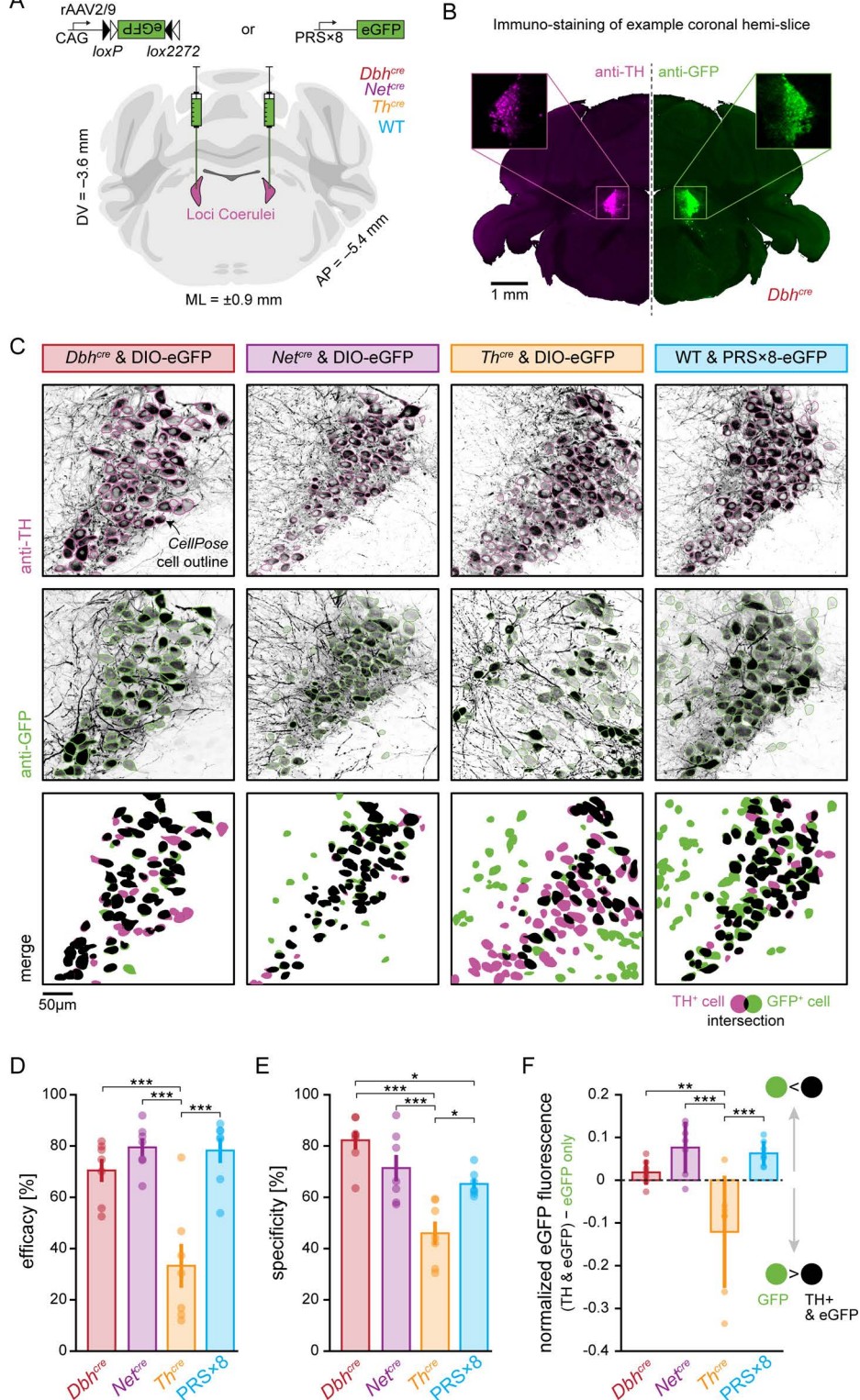

**Fig 1. Viral transduction of LC-NE neurons in different model systems.** (**A**) Either CAG-DIO-eGFP (in *Dbh^cre^*, *Net^cre^*, and *Th^cre^*) or PRS×8-eGFP (in WT) was bilaterally injected into the LC of mice (*n* = 7 animals in each group). Coronal brain slice image from [37]. (**B**) Example LC hemi-brain slice (vertically mirrored) from a *Dbh^cre^* animal. Noradrenergic neurons were identified by immunostaining against TH (left, magenta), while transgene

expression was visualized with an anti-GFP staining (right, green). (**C**) Example LC cells in each of the four model systems (columns) immuno-stained against TH (first row, magenta outlines) and GFP (second row, green outlines). Cells were identified with *CellPose* and boundaries are denoted with colored outlines. Resulting cell masks were then overlaid (third row), and each $TH^+$ cell was labeled as $GFP^+$ (successful transduction, black) when cell masks overlapped 50% or more the size of the $TH^+$ cell, or $GFP^-$ otherwise (missed transduction, magenta). $GFP^+$ and $TH^-$ (erroneous transduction, green) was counted as unspecific transduction. (**D**) Efficacy (i.e., the true positive rate), computed as $N(GFP^+|TH^+)/N(TH^+)$, across model systems. (**E**) Specificity (i.e., the inverse of the false positive rate), computed as $N(TH^+|GFP^+)/N(GFP^+)$, across model systems. (**F**) Difference in normalized eGFP fluorescence of neurons that co-expressed eGFP and TH (i.e., true positive neurons) and neurons that expressed eGFP only (i.e., false positive neurons) across model systems. Negative values indicate that eGFP fluorescence of TH-negative neurons was brighter than eGFP fluorescence of TH-expressing neurons. In (**D–F**) error bars denote the standard error of the mean (with $n = 7$), while statistical significance is denoted by */**/*** for $p < 0.05/0.01/0.001$, respectively. Only significant differences are indicated. Numerical data underlying panels D–F can be found in S1 Data.

and PRS×8 mice (0.02 ± 0.03/ 0.08 ± 0.06/0.06 ± 0.03, respectively), it was lower than the eGFP fluorescence of true positive cells in *Th^cre* mice (−0.12 ± 0.13), indicating that the levels of cre-mediated transgene expression are functionally de-coupled from the levels of native TH expression in this mouse line (one-way ANOVA, $F_{24} = 10.1$, $p = 0.0002$; Fig 1F). Similarly, at the cellular level, normalized eGFP fluorescence in true positive neurons was higher than in false positive neurons in *Dbh^cre*, *Net^cre,* and PRS×8 mice, while eGFP fluorescence in false positive neurons exceeded the fluorescence of true positive neurons in *Th^cre* mice ($p < 0.1 × 10^{-5}$ for all groups, Bonferroni-corrected Wilcoxon rank sum test; S2 Fig). These findings indicate that cre recombinase activity and endogenous TH expression are decoupled in *Th^cre* mice, and further corroborate the heterogeneous expression patterns across approaches.

## Control experiments

We performed a number of controls to validate our results and to rule out potential sources of confounding.

**Effects of overlap threshold.** In our analyses, we have chosen an overlap threshold between $eGFP^+$ and $TH^+$ cells of ≥50% to define co-expression. To assess the generality of our results, we systematically varied the threshold from 5% to 95% (in 5% increments). Relative differences in both efficacy and specificity were independent of the overlap threshold in this range (S3 Fig).

**Effects of leaky virus expression.** One factor that may lead to underestimation of transduction specificity in cre lines is non-specific transgene expression of cre-dependent constructs in cre-negative cells, as previously described [38]. To assess the extent of this effect, we bilaterally injected the same rAAV2/9-CAG-DIO-eGFP in the LC of 2 wild-type mice. We could not detect eGFP-expressing cells in these animals, ruling out leaky expression of our viral constructs at the titers used in this study (S4 Fig).

**Effects of cell segmentation errors.** To account for potential segmentation errors, we compared results obtained from *CellPose* with manual cell identification by two independent experimenters (CW and AD) in a subset of slices and found agreement between *CellPose* and experimenter rating, as well as between experimenters (S5 Fig). Of note, the training data for *CellPose* was annotated by a third experimenter (LSE), which eliminated experimenter effects. To further validate our experimental routine and our analysis pipeline, we injected rAAV2/9-CAG-DIO-eGFP also into the ventral tegmental area (VTA) of *Th^cre* mice (S6A and S6B Fig). In the VTA, we found a specificity of 60.7 ± 6.8% (S6C and S6D Fig), which is in line with the specificity reported by Lammel and colleagues [39] (59 ± 1%; based on visual inspection), and further confirms that our automated analysis is comparable to manual curation by a human observer.

**Effects of GFP staining.** To investigate the role of GFP staining in our findings, we used a secondary antibody in the red spectrum against our primary anti-GFP staining and directly compared cells detected in the green (native eGFP) and red (GFP-stained) channels in a subset of animals (S7A Fig). We found that the anti-GFP staining increased the number of cells detected by *CellPose* by 21.5 ± 15.3% (*t* test against 0: $t_{14} = 4.8$, $p = 2.8 × 10^{-4}$; S7B Fig), and that the averaged normalized brightness of stained cells exceeded the one of native eGFP-expressing cells by 55.0 ± 35% (*t* test against 0: $t_{14} = 6.66$, $p = 1.08 × 10^{-5}$; S7C Fig). This change in fluorescence was significantly stronger in the dimmest eGFP-expressing

cells, i.e., cells expressing relatively low levels of eGFP ($t$ test against 1: $t_{12} = -11.56$, $p = 1.5 \times 10^{-8}$; S7D and S7E Fig). Unfortunately, it is unclear whether low levels of native eGFP expression are due to low promoter activity or weak viral transduction. If low promoter activity causes low eGFP levels, the transgene should accumulate over time, as even low levels of cre recombinase will eventually recombine all floxed genes. Hence, the GFP staining would compensate for the number of detected cells. In contrast, if low eGFP levels result from differences in viral transduction of individual cells (i.e., directly reflects the number of transgene copies per cell) the GFP staining would inflate the number of detected cells. Of note, the changes in fluorescence and cell detection introduced by the anti-GFP staining apply to all experimental groups in the same way.

### Ectopic expression of transgenes

In addition to differences in the specificity of neural transduction within the LC and its immediate surroundings, we also observed ectopic eGFP expression in various brain regions more distant from the LC (Fig 2). Again, we found differences between the different model systems with respect to ectopic LC transgene expression. The lowest ectopic expression was found in *Dbh*$^{cre}$ animals, where we observed eGFP expression only in small numbers of cells in the cerebellum (<10 neurons, 3/7 animals; S8A Fig) and in the inferior colliculus (<10 neurons, 1/7 animals; Fig 2A). In *Net*$^{cre}$ mice, we found a small number of eGFP-expressing cells in the cerebellum (<10 neurons, 4/7 mice Fig 2B), and moderate eGFP expression in the vestibular nucleus of a single mouse (~10–100 neurons, S8B Fig). In *Th*$^{cre}$ mice, we found substantial ectopic expression in the lateral parabrachial nucleus (~10–500 neurons, 7/7 mice; S8C Fig), moderate expression in the inferior colliculus (10–100 neurons, 2/7 mice; Fig 2C) and central gray (<100 neurons, 1/7 mice; S8D Fig) and low expression in

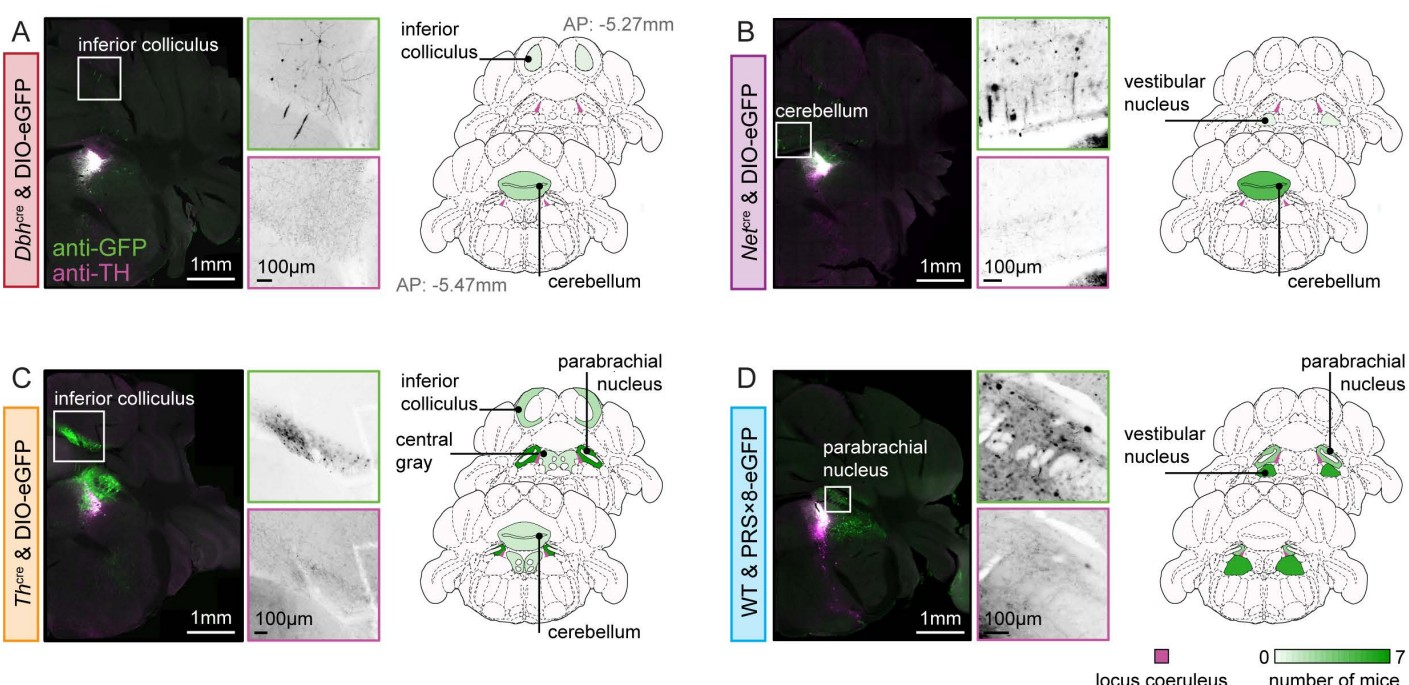

**Fig 2. Ectopic transgene expression across model systems. (A–D)** On the left panels: example brain slices showing ectopic expression in *Dbh*$^{cre}$/*Net*$^{cre}$/*Th*$^{cre}$/PRS×8 model systems. Immuno-staining against GFP and TH is displayed in green and magenta, respectively. On the right panels: summary of ectopic expression across all mice in *Dbh*$^{cre}$/*Net*$^{cre}$/*Th*$^{cre}$/PRS×8 model systems. Brain regions are color-coded to reflect the number of mice which showed ectopic expression (at least 1 neuron) in the corresponding brain region. Brain slice images from [37].

the cerebellum (<10 neurons, 1/7 mice; S8E Fig). Finally, in PRS×8 animals, we found moderate eGFP expression both in the vestibular nucleus (~10–200 neurons, 6/7 animals; S8F Fig), and in the lateral parabrachial nucleus (~10–100 neurons, 3/7 animals; Fig 2D).

**Effects of promoter, serotype, and injection volume on transduction of LC-NE neurons**

Next, we systematically varied the promoter controlling transgene expression, the serotype of the rAAV, and the volume of injected virus suspension, in order to investigate potential effects of these parameters on the transduction of LC-NE neurons.

**Effects of promoter.** We first assessed whether the choice of the promoter had an influence on the efficacy and specificity of transgene expression in LC-NE neurons. The human synapsin (hSyn) promoter [40,41] is a relatively small promoter, which has been widely used in neuroscientific research, and it facilitates transgene expression broadly across all neuronal subtypes. Thus, we injected an additional set of *Dbh^cre^* mice (*n*=7) with rAAV2/9 encoding eGFP in a cre-dependent manner under control of the hSyn promoter (rAAV2/9-hSyn-DIO-eGFP; Fig 3A). We observed robust eGFP expression in all injected LC (Fig 3B), and could not detect significant differences between *Dbh^cre^* mice injected with rAAV2/9-hSyn-DIO-eGFP and rAAV2/9-CAG-DIO-eGFP, neither in terms of efficacy (78.5±4.3 versus 70.5±11.8, *p*=0.12, two-sample *t* test; Fig 3C) nor in terms of specificity (83.5±10.5 versus 82.2±9.5, *p*=0.82, two-sample *t* test; Fig 3D). Hence, our findings are not restricted to the CAG promoter, and reliable transgene expression in LC-NE cells of *Dbh^cre^* mice can also be achieved under the control of hSyn.

**Effects of viral serotype.** Since transgene expression in *Th^cre^* mice, and to a lower degree also under the PRS×8 promoter, resulted in ectopic eGFP expression in regions distant to the LC, we suggest that less viral spread can help to reduce ectopic transgene expression. It was previously shown in cortical tissue that rAAV2/2 shows less spread compared to the serotypes rAAV2/1, rAAV2/5, rAAV2/8, and rAAV2/9 [42]. To get a precise understanding of serotype effects on viral spread, we injected wild-type mice either with rAAV2/9 or rAAV2/2 encoding eGFP unconditionally under control of the hSyn promoter (Figs 3E and 3F and S9). The viral spread of transgene expression was then quantified by smoothing, normalizing, and thresholding of immuno-stained hemi-brain slices. Indeed, we found that transgene expression upon viral transduction using rAAV2/2 led to a more restricted spread as compared to rAAV2/9, at least when thresholding fluorescence up to 0.5 of its maximum value (i.e., when analyzing regions relatively distant to the injection site; *p*=0.0002/0.0002/0.0003/0.0033/0.028 for thresholds of 0.1/0.2/0.3/0.4/0.5, respectively; two-sample *t* test, *n*=6 LC/serotype; Fig 3G).

**Effects of injection volume.** As an alternative to using a different viral serotype, we aimed to reduce viral spread by lowering the injection volume. In these experiments, we bilaterally injected either 300 nl (the standard volume used in other experiments in this manuscript), 100 nl, or 50 nl of rAAV2/9-hSyn-eGFP into the LC of wild-type mice (Figs 3H and 3I and S9) and quantified the viral spread as described above. Significant differences in viral spread were observed for thresholds up to 50% of the maximum fluorescence (one-way ANOVA, $F_{15}$ = 5.3/13.9/13.6/8.63/4.38, *p*=0.02/0.0004/0.0004/0.003/0.03 for thresholds of 0.1/0.2/0.3/0.4/0.5, respectively; Fig 3J). More specifically, the viral spread upon injection of 100 nl of viral suspension was significantly lower as compared to the spread of 300 nl for thresholds up to 0.4 (*p*=0.03/0.002/0.007/0.04, Tukey's test), while the spread upon injection of 50 nl was significantly lower as compared to 300 nl for thresholds up to 0.5 (*p*=0.04/0.0005/0.0004/0.003/0.03, Tukey's test). No significant differences between the viral spread following 50 nl and 100 nl injections were found at any threshold.

These experiments demonstrate that the viral spread observed upon LC injections can be reduced both by implementing serotype rAAV2/2 and by lowering the injection volume of viral suspension. Hence, the use of serotype rAAV2/2 or lower injection volumes might help to reduce ectopic expression in model systems that showed ectopic expression in brain regions distant to the LC (i.e., *Th^cre^* and, to a milder degree, PRS×8)

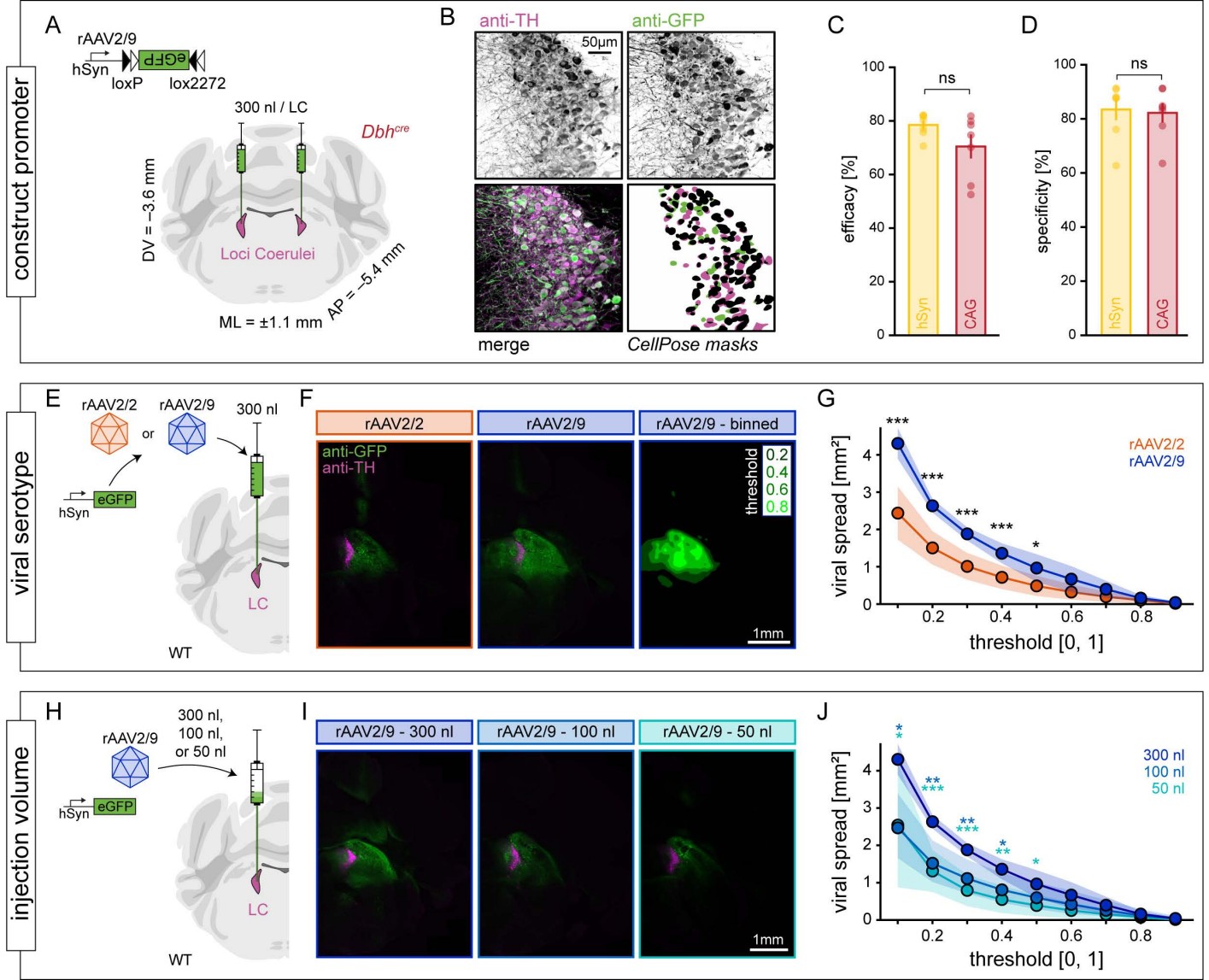

**Fig 3. Effects of promoter, serotype, and injection volume on viral transduction of LC-NE neurons.** (**A**) To investigate effects of the promoter on viral transduction, rAAV2/9-hSyn-DIO-eGFP (a construct analog to rAAV2/9-CAG-DIO-eGFP used in Figs 1 and 2) was bilaterally injected into the LC of *Dbh^cre* mice (*n* = 7 animals). Coronal brain slice image from [37]. (**B**) Example LC section from an rAAV2/9-hSyn-DIO-eGFP-injected *Dbh^cre* mouse immuno-stained against TH (top left) and GFP (top right), as well as color-coded overlay (bottom left). Cells were identified with *CellPose*. (**C**) Efficacy ($N$(GFP$^+$|TH$^+$)/$N$(TH$^+$)) and (**D**) Specificity ($N$(TH$^+$|GFP$^+$)/$N$(GFP$^+$)) were comparable upon injection of hSyn-DIO-eGFP (yellow) or CAG-DIO-eGFP (red; re-plotted from Figs 1D and 2) in *Dbh^cre* mice (ns = not significant; two-sample *t* test, *n* = 7 each). (**E**) To investigate effects of the viral serotype on viral transduction, 300 nl of either rAAV2/2-hSyn-eGFP or rAAV2/9-hSyn-eGFP were bilaterally injected into the LC of wild-type mice (*n* = 6 LC from *n* = 3 mice each). (**F**) Example images of hemi-brain slices of an rAAV2/2-injected (orange, left) and an rAAV2/9-injected mouse (blue, center). Images were smoothed and fluorescence was normalized, before thresholding images with an increment of 0.1 and calculating the area above threshold (see example with an increment of 0.2 on the right). (**G**) Viral spread was quantified at various thresholds, revealing a more restricted spread upon injection of rAAV2/2 as compared to rAAV2/9 (*/**/*** for *p* < 0.05/0.01/0.001, two-sample *t*-tests, *n* = 6 LC per group). (**H**) To investigate effects of the injection volume on viral transduction, either 300, 100, or 50 nl of rAAV2/9-hSyn-eGFP were bilaterally injected into the LC of wild-type mice (*n* = 6 LC from *n* = 3 mice each). (**I**) Example images of hemi-brain slices of mice injected with 300 nl (dark blue, left), 100 nl (light blue, center) or 50 nl of rAAV2/9-hSyn-eGFP (cyan, right). (**J**) Viral spread was quantified at various thresholds, revealing a more restricted spread upon injecting 100 and 50 nl of viral suspension as compared to 300 nl of viral suspension (*/**/*** for *p* < 0.05/0.01/0.001, one-way ANOVA and Tukey's test, *N* = 6 LC per group). Asterisks denote significance between the 300 nl group and the 100 nl/50 nl group, respectively, as no significant differences were found between the 100 nl and the 50 nl group. Numerical data underlying panels C, D, G, and J can be found in S1 Data.

## Success rates of LC targeting

Occasionally, we observed complete absence of eGFP (i.e., transgene expression) in one of the two LC of bilaterally injected animals, while expression was strong in the second LC. This pattern of unilateral transgene expression was found in 3/2/1/3 out of each 5 *Dbh^cre^*/*Net^cre^*/*Th^cre^*/PRS×8 mice, respectively (Fig 4A). Since we used the same micropipette to inject the viral suspension sequentially in both LC, we were concerned that this approach might have compromised the success rate of injections (e.g., due to possible inhomogeneous distribution of viral particles in the suspension). Therefore, we added two more mice to each group using a new micropipette for each LC. However, also here, we observed unilateral transgene expression in some cases (0/0/2/1 *Dbh^cre^*/*Net^cre^*/*Th^cre^*/PRS×8; Fig 4B), which did not statistically differ from the previous group where the same pipette was used ($p = 0.72$, chi-squared test, $n = 20/8$ mice; Fig 4C). Thus, sequential injection with the same pipette did not explain the occasional failure of virus transduction. Importantly, infusion of the viral suspension was also visually confirmed during each injection, and hence the absence of expression in some LC could not be attributed to clogged micropipettes.

The overall success rate of injections in our study (80%, 59/74 LC) is in line with previous reports (76%–80%) [43,44], which attributed the lack of transgene expression to off-target virus administration (i.e., injection in brain regions adjacent to LC). To test this possibility, we injected *Dbh^cre^*, *Net^cre^*, and *Th^cre^* mice ($n = 3$ each) with a so-called 'cre-switch' construct which consists of a DIO for tdTomato and an inverted open reading frame for eGFP (DO-tdTomato-DIO-eGFP) [45]. Hence, cell-type specific expression of eGFP in cre-positive cells allows for characterization of the molecular specificity, while expression of tdTomato in cre-negative cells allows for estimation of the viral spread (and potential off-target virus administration) at the injection site (Fig 4D). Four weeks after injection, expression of tdTomato in cre-negative cells revealed a broad viral spread in most animals, reaching from the midline to the parabrachial nucleus in the mediolateral axis, and from the fourth ventricle to the central gray in the dorsoventral axis (Fig 4E). However, also in these experiments we observed three animals (out of nine) with unilateral eGFP expression where also tdTomato was absent in the non-expressing hemisphere (Fig 4F), suggesting that absence of expression in LC was not due to off-target injection in neighboring brain tissue but rather a failure of virus infection in general.

We therefore suggest that the lack of expression in some LC might result from off-target administration into the 4th ventricle, which is in close proximity to the LC. This mistargeting could result from minor misalignment on the mediolateral axis during virus injection, resulting in more lateral virus injection in one hemisphere, but virus injection into the 4th ventricle in the other hemisphere. Indeed, when changing the injection coordinates from ±0.9 mm to ±1.1 mm lateral relative to bregma in subsequent experiments (i.e., injection of AAV2/9-hSyn-DIO-eGFP in *Dbh^cre^* mice and injection of AAV2/2 or AAV2/9-hSyn-eGFP in WT mice; see Figs 3 and S9), we observed bilateral transgene expression in 7/7 and 12/12 LC (Fig 4F and 4G). These success rates are significantly higher than the ones observed when injecting ±0.9 mm lateral relative to bregma (Fig 4H and 4I; $p = 0.032$ for conditional expression, chi-squared test, $n = 28/7$ mice; $p = 0.031$ for unconditional expression, chi-squared test, $n = 9/12$ mice). Thus, more laterally targeted injections lead to increased transduction rates of the LC, likely by lowering the risk of injecting the virus suspension into the 4th ventricle.

## Behavioral screening of cre lines

Unlike the PRS×8 approach, cre-dependent transgene expression relies on the use of mouse lines expressing cre recombinase in a molecularly defined set of cells. As phenotypic alterations have been broadly reported for neuromodulatory cre driver lines [46–48], we screened the driver lines used in this study for behavioral alterations. We applied a series of well-established behavioral paradigms to quantify spontaneous locomotion, anxiety, exploratory behavior, working memory, and spatial learning and memory.

As LC-NE activation is strongly linked to stress and anxiety [6,49], we first compared anxiety-like behavior in cre-expressing *Dbh^cre^*, *Net^cre^*, and *Th^cre^* mice to their wild-type littermates. To this end, mice were left to freely explore an open

PLOS Biology

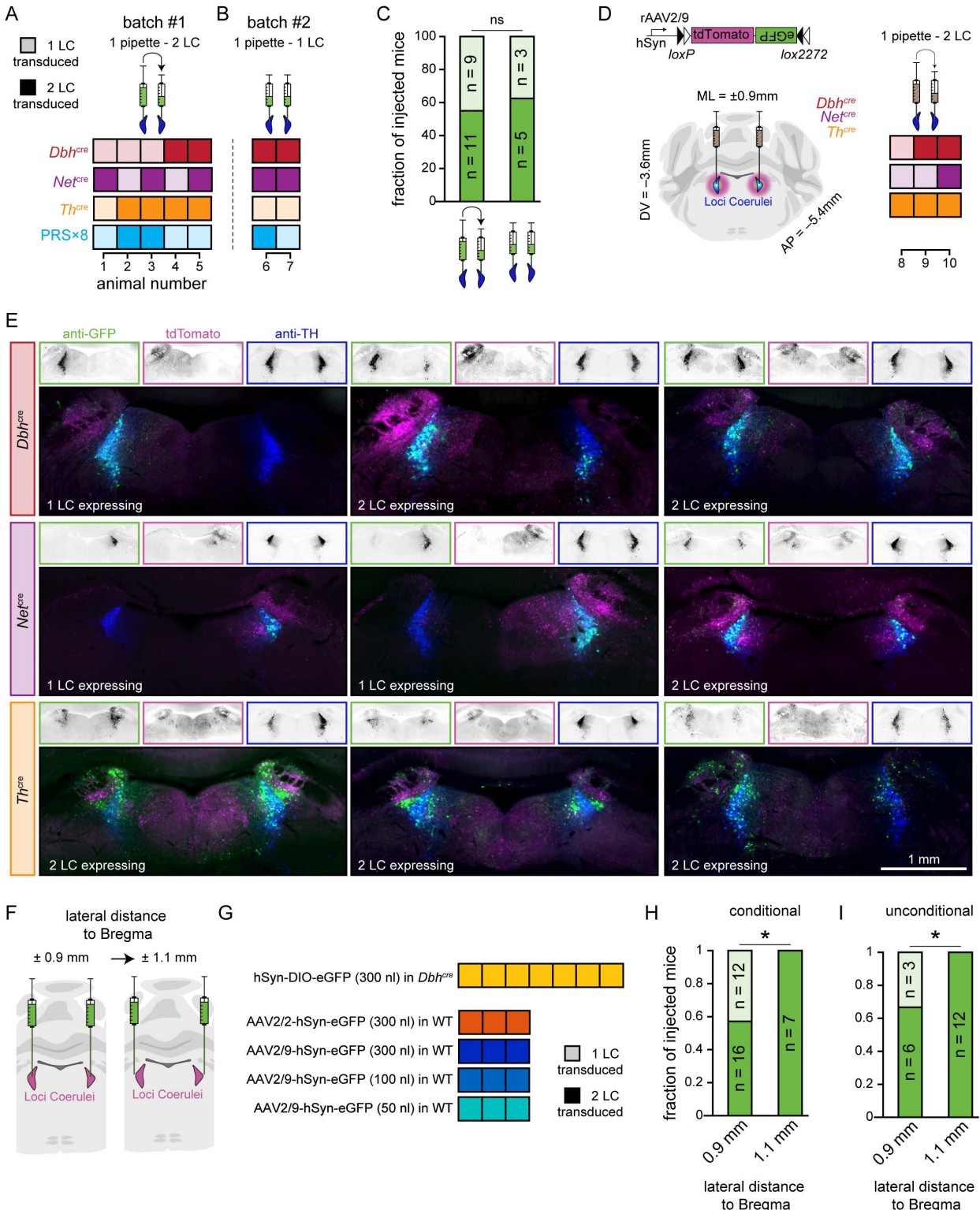

**Fig 4. Improving LC targeting.** (**A**) Bilateral transduction efficacy (*n* = 5 animals in each group) using a single micropipette to sequentially infect both LC (2 × 300 nl). (**B**) Same as A, except that one micropipette was used for each LC (*n* = 2 animals in each group). (**C**) Fraction of animals with bilateral (solid) or unilateral (shaded) LC transduction upon LC injections with a single pipette vs. two pipettes. (**D**) Assessement of viral spread by injections of a

'cre-switch' construct leading to eGFP expression in a cre-dependent manner and tdTomato expression in cre-negative cells [45]. (**E**) Transgene expression in *Dbh^cre^*, *Net^cre^* and *Th^cre^* animals (*n* = 3 animals each) shows conditional eGFP expression in cre-positive cells (green) and tdTomato expression in cre-negative cells (magenta). LC-NE cells are visualized by immunostaining against TH (dark blue). Importantly, in animals without eGFP expression, cre-negative cells around the LC also lacked the expression of tdTomato, likely resulting from loss of the virus suspension to the ventricle. (**F**) Change of mediolateral coordinates for virus injections from ±0.9 mm to ±1.1 mm. (**G**) Bilateral transduction efficacy after using the new coordinates in transgenic or wild-type animals. (**H**) Fraction of conditionally eGFP-expressing animals with bilateral (solid) or unilateral (shaded) LC transduction upon LC injections at mediolateral coordinates of from ± 0.9 mm and ± 1.1 mm (e.g., pooled data from panel A/B vs. panel G, uppermost row). (**I**) Fraction of unconditionally fluorophore-expressing animals with bilateral (solid) or unilateral (shaded) LC transduction upon LC injections at mediolateral coordinates of ±0.9 mm and ±1.1 mm (e.g., pooled data from panel D vs. panel G, rows 2–5). ns = not significant, * for *p* < 0.05, chi-squared test.

field arena for 20 min (Fig 5A, left) [50]. As mice show the intrinsic tendency to avoid open areas, the time spent near the walls of the arena serves as a proxy for anxiety, with less anxious mice spending more time in the center. While not all groups displayed significant avoidance of the arena center ($p = 0.004/0.06/4 \times 10^{-4}/ 4 \times 10^{-5}/0.14/6 \times 10^{-5}$ for *Dbh^cre^*/*Dbh^WT^*/*Net^cre^*/*Net^WT^*/*Th^cre^*/*Th^WT^*, respectively; one-sample *t* test against chance level (16%), *n* = 16 mice per group), anxiogenic conditions of the paradigm could be confirmed when pooling the data of all mice ($p = 6 \times 10^{-15}$, *n* = 96 mice). When comparing eight males and eight females of each heterozygous cre-expressing line with a corresponding number of wild-type littermates of each mouse line, we did not detect any significant behavioral difference that could be explained by either genotype, sex, or the interactions of these two factors in any mouse line (Fig 5A, right; two-way ANOVA; genotype effects: *Dbh^cre^*: $F_{28} = 0.45$, $p = 0.51$; *Net^cre^*: $F_{28} = 0.54$, $p = 0.47$; *Th^cre^*: $F_{28} = 0.01$, $p = 0.91$; effects of sex and interactions of sex and genotype are shown in S2 Table). Also, additional readouts used to approximate anxiety and spontaneous locomotion did not indicate any genotype-dependent behavioral difference in any mouse line (S10 Fig).

As an additional experiment approximating anxiety-related behavior, mice were left to explore an elevated plus maze (EPM) [51] for 5 min. The EPM consists of four arms radiating from a central platform, with two opposing arms being walled and the other two arms being open (Fig 5B, left). Thus, mice need to balance their natural exploratory drive with their intrinsic tendency to avoid exposed areas. At the start of the experiment, mice were placed on the central platform, and the time spent in the open and closed arms served as a proxy for anxiety, with less anxious mice spending more time in the open arms. Again, significant avoidance of the open arms could not be demonstrated in each experimental group ($p = 0.08/0.002/5 \times 10^{-4}/1 \times 10^{-4}/0.01/0.23$ for *Dbh^cre^*/*Dbh^WT^*/*Net^cre^*/*Net^WT^*/*Th^cre^*/*Th^WT^*, respectively; one-sample *t* test against chance level (48%), *n* = 16 mice per group), but the anxiogenic nature of this paradigm could be validated when pooling the data of all mice ($p = 3 \times 10^{-12}$, *n* = 96 mice). Also, in the EPM, we did not detect any behavioral difference that could be explained by either genotype, sex, or the interactions of these two factors in any mouse line (Fig 5B, right; two-way ANOVA; genotype effects: *Dbh^cre^*: $F_{28} = 1.96$, $p = 0.17$; *Net^cre^*: $F_{28} = 0.01$, $p = 0.93$; *Th^cre^*: $F_{28} = 0.32$, $p = 0.57$; effects of sex and interactions of sex and genotype are shown in S2 Table). Similarly, no behavioral alterations were found in any additional parameters extracted from the EPM (S11 Fig). In conclusion, we did not detect any genotype-related changes in anxiety-like behavior in any mouse line tested in two distinct experimental paradigms under light anxiogenic conditions.

Recent studies have demonstrated the involvement of the LC-NE system in the formation of episodic and spatial memory [52,53], which we investigated in two additional experiments. First, we employed the Y-maze paradigm to approximate working memory in mice. In this assay, mice are placed in a maze with three symmetrically radiating arms, which they are free to explore. Based on the innate tendency to explore novel environments, mice spontaneously alternate between arms, i.e., when leaving one arm, they tend to move to the arm that was not previously visited, rather than the arm that was previously explored (Fig 5C, left; visual cues were given to distinguish the different arms) [54,55]. This was quantified as the fraction of alternations (i.e., consecutive visits of all three arms) over the total number of transitions. The fraction of alternations was above chance level in all groups ($p < 0.01$; Bonferroni-corrected one-sample *t* test), indicating intact working memory in all groups. No genotype-related behavioral changes were found (Fig 5C, right; two-way ANOVA; genotype

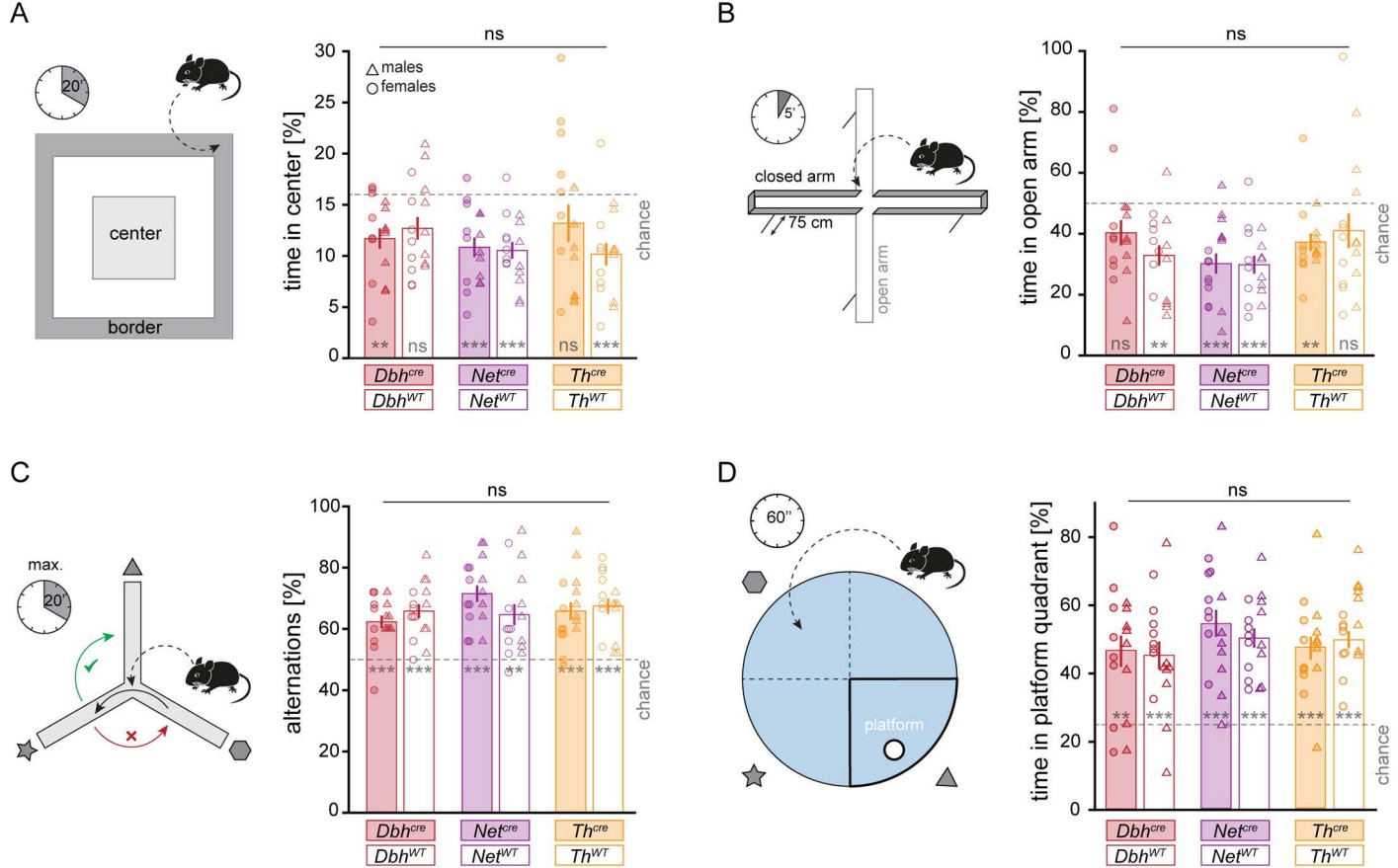

**Fig 5. Behavioral screening of cre-expressing mice.** (**A**) Left: To assess anxiety-like behavior, mice were placed in one corner of the arena and left to explore the open field arena for 20 min. Right: The percentage of time spent in the center of the arena indicates no difference in anxiety-like behavior of cre-expressing mice (filled bars) as compared to wild-type littermates (empty bars) of different driver lines. (**B**) Left: To further test anxiety-like behavior, mice were placed on the central platform of the elevated plus maze and left to explore the maze for 5 min. Right: The percentage of time spent in open arms indicates no difference in anxiety-like behavior of cre-expressing mice as compared to wild-type littermates in any driver line. (**C**) Left: To assess working memory, mice were placed in the center of the Y-maze and left to explore the maze until 26 transitions were made, for a maximum of 15 min. Upon the first transition between arms (black arrow), the next transition is expected to occur towards the previously unexplored arm (green arrow), rather than to the arm that was just visited before (red arrow). Right: The percentage of alterations (i.e., consecutive visits of all three arms) over the total number of transitions is above chance level (dashed line) in all groups, while no genotype-dependent differences in working memory were detected between cre-expressing mice and wild-type littermates in any mouse line. (**D**) Left: To test spatial memory, mice are trained to find a hidden platform in a circular pool filled with opaque water on two consecutive days. On day three, the platform is removed, and the time spent in the area where the platform was previously located serves as a proxy for spatial memory. Right: the percentage of time spent in the platform quadrant was above chance level (dashed line) in all groups, while no genotype-dependent differences in spatial memory were detected between cre-expressing mice and wild-type littermates in any mouse line. All data is depicted as mean ± standard error of the mean. ns = not significant. **/*** for $p < 0.01/0.001$, respectively, tested against chance level. $n = 8$ females (circles) and 8 males (triangles). Numerical data underlying this figure can be found in S1 Data.

effects: $Dbh^{cre}$: $F_{28} = 1.23$, $p = 0.26$; $Net^{cre}$: $F_{28} = 2.48$, $p = 0.13$; $Th^{cre}$: $F_{28} = 0.23$, $p = 0.63$; effects of sex and interactions of sex and genotype are shown in S2 Table, additional results from Y-maze experiments are shown in S12 Fig).

Finally, to investigate spatial memory, we performed the Morris Water Maze (MWM) task [56]. In the MWM, mice are placed in a circular pool filled with opaque water that contains a hidden platform placed just below the surface that mice can access to escape the water (Fig 5D, left; visual cues for spatial orientation were placed around the pool). The MWM task was performed on three consecutive days after habituating mice to the water. On the first two days, each mouse

was placed in the water four times, at pseudo-randomized positions in each trial. Decreased latencies and travel distances to reach the platform during these training sessions indicated successful spatial learning in all experimental groups (S13 Fig). On the second day, after the first two training trials, a probe trial was performed, in which the platform was removed. The time spent in the platform quadrant approximates short-term memory. On day three, a single probe trial was done, from which long-term memory can be measured. The time spent in the platform quadrant during the probe trial on day 3 was above chance level in all groups ($p < 0.01$; Bonferroni-corrected one-sample $t$ test), indicating intact spatial memory formation. No genotype-dependent behavioral changes were found (Fig 5D, right; two-way ANOVA; genotype effects: $Dbh^{cre}$: $F_{28} = 1.14$, $p = 0.29$; $Net^{cre}$: $F_{28} = 0.56$, $p = 0.46$; $Th^{cre}$: $F_{28} = 0.34$, $p = 0.56$; effects of sex and interactions of sex and genotype, as well as results of the short-term memory test on day two are shown in S2 Table, additional results of the MWM experiments on day 3 are shown in S14 Fig). In conclusion, we found no genotype-related behavioral differences between cre-expressing mice and wild-type littermates in $Dbh^{cre}$, $Net^{cre}$, or $Th^{cre}$ mice in the four behavioral assays considered.

## PRS×8-based constructs for monitoring and manipulating LC activity

Transgene expression driven by the PRS×8 promoter, which has high efficacy and good local specificity (Fig 1), is independent of cre recombinase. Thus, molecular targeting of LC-NE neurons with the PRS×8 promoter can be applied either to wild-type mice, eliminating the need for a cre driver line, or to a different cre driver line, providing access to two genetically distinct neuronal populations in the same animal [32]. In addition, PRS×8-based transgene expression can be applied to different species, as demonstrated in rats [25,28,57,58]. As there are currently very few AAV-carried PRS×8-based constructs available, we developed a set of PRS×8-based constructs to enable the monitoring and control of LC-NE activity (of note, canine Adenovirus type 2 (CAV-2) carrying PRS×8-mediated transgenes are commercially available at https://plateau-igmm.pvm.cnrs.fr/?cat=69).

To optically monitor LC-NE activity, we developed a construct encoding jGCaMP8m [59] under the control of the PRS×8 promoter. We directly compared its functionality to a conditionally expressed indicator in the contralateral LC of the same animal. To do so, we injected rAAV2/9-PRS×8-jGCaMP8m into one LC of a $Dbh^{cre}$ mouse and a cre-dependent, spectrally compatible calcium indicator jRGECO1a [60] (AAV2/1-hSyn-DIO-jRGECO1a) into the contralateral LC (Fig 6A and 6B). In line with our previous results (Fig 1), we observed a specificity of 92.5% and an efficacy of 79.6% for PRS×8-jGCaMP8m, and a specificity and efficacy of 94.9% and 64.3% for hSyn-DIO-jRGECO1a. We then implanted optical fibers above the injection sites and obtained fiber photometry recordings simultaneously from both LC of awake, head-fixed mice, while monitoring the pupil diameter of the animals (Fig 6C). We observed strong correlations between the neuronal dynamics of jGCaMP8m- and jRGECO1a-expressing LC ($r = 0.73$, $p > 0.0001$; Pearson's correlation coefficient for a recording of 20 min duration). Since the LC-NE system causally controls pupil-linked arousal [61], we aligned all signals to local peaks of the derivative of the pupil diameter that occurred in the absence of locomotion, as pupil dynamics and locomotion are tightly coupled in mice [62]. As expected, peaks in pupil size were preceded by peaks in LC activity, which confirms the biological relevance of the recorded signals (Fig 6D). Importantly, similar temporal dynamics of LC activity were observed in both channels, indicating that PRS×8- and $Dbh^{cre}$-mediated expression of calcium indicators reveals comparable functional signals of LC activity.

Importantly, when exciting the calcium indicators at their quasi-isosbestic wavelength (405 nm), no peaks were observed in either channel (Fig 6E), confirming that the observed peaks in fluorescence indeed resulted from calcium dynamics, and not from motion or hemodynamic artifacts. Additional controls were performed to exclude spectral crosstalk of the calcium indicators by restricting our recording from both LC either to the green or to the red spectral range. When aligning signals to pupil dilation, the previously observed peak in jRGECO1a fluorescence was not present in the green spectrum (Fig 6F), and, vice versa, the previously observed peak in PRS×8-jGCaMP8m fluorescence was abolished in the red spectral range (Fig 6G), confirming absence of spectral crosstalk between the two indicators.

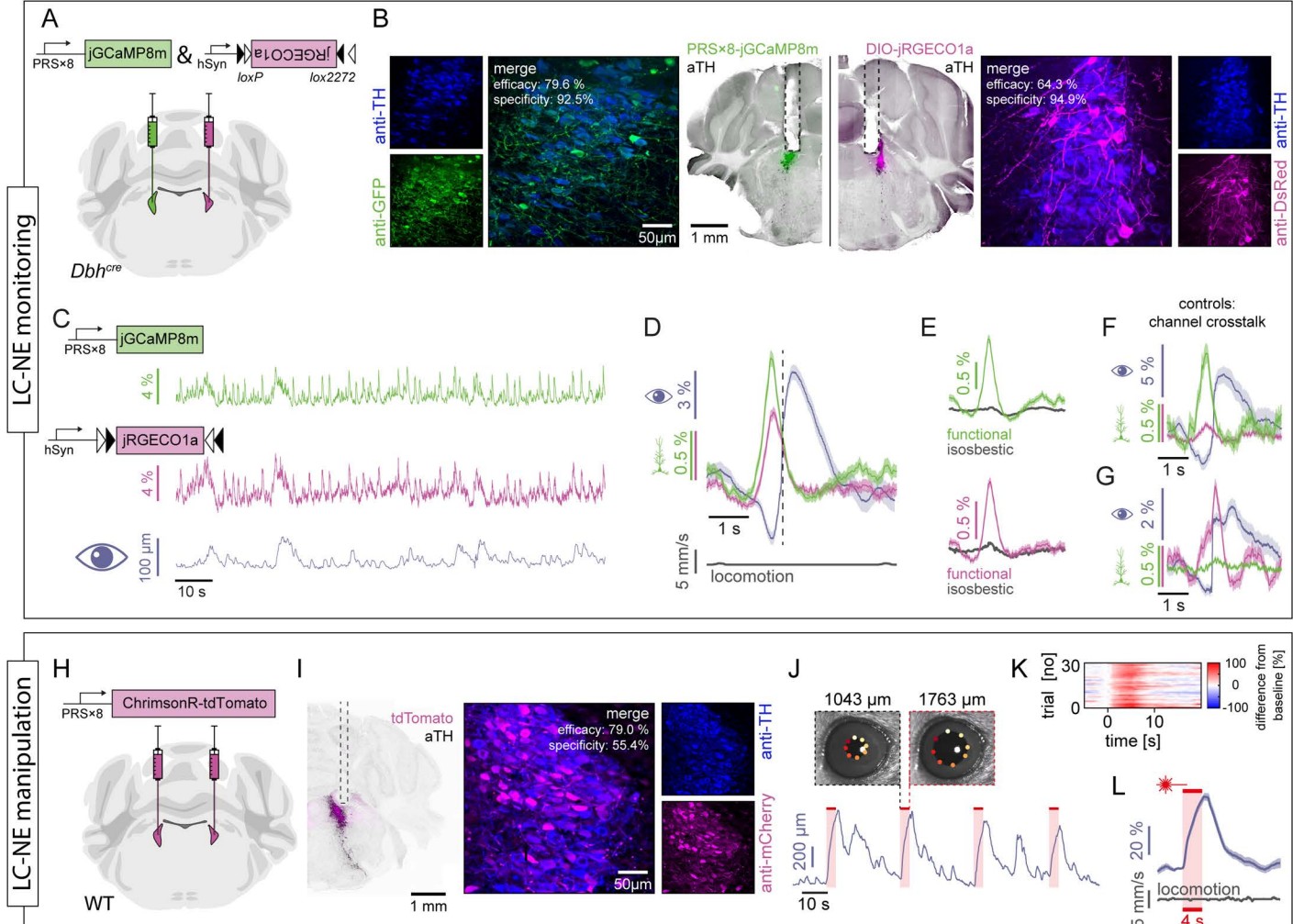

**Fig 6. PRS×8-mediated constructs to monitor and manipulate LC activity.** (**A**) PRS×8-jGCaMP8m was injected into one LC of a *Dbh^cre* mouse and hSyn-DIO-jRGECO1a was injected in the contralateral LC. (**B**) Epifluorescence images of hemi-brain slices expressing PRS×8-driven jGCaMP8m (green, left) and cre-dependent jRGECO1a (magenta, right). Immunostaining against TH is shown in black, while dashed lines indicate the optical fiber tracts. Confocal images of anti-TH and anti-GFP staining, as well as a merged image with quantification of efficiency and specificity are shown next to the overview images. (**C**) Simultaneous fiber photometry recordings of jGCaMP8m fluorescence (top) and jRGECO1a fluorescence (center), along with the pupil diameter of the mouse (bottom). (**D**) ΔF (mean ± standard error of the mean) of jGCaMP8m (green) and jRGECO1a (magenta) as well as pupil diameter (gray) locked to local peaks in the derivative of the pupil diameter (indicated by dashed line, *n* = 171 peaks from 20 min of recording). Only events in the absence of locomotion (gray, bottom) were analyzed. (**E**) Pupil-aligned functional and isosbestic fluorescence of jGCaMP8m (top) and jRGECO1a (bottom). Functional excitation was done at 470 nm for jGCaMP8m and 560 nm for jRGECO1a, while isosbestic excitation for both indicators was done at 405 nm. (**F**) ΔF of green-light-filtered GCaMP8m (green) and jRGECO1a (magenta), locked to peaks in the derivative of pupil diameter, exclude major cross-talk from jGCaMP8m to the jRGECO1a channel (*n* = 52 peaks). (**G**) ΔF of red-light-filtered jGCaMP8m (green) and jRGECO1a (magenta), locked to peaks in the derivative of pupil diameter, exclude major cross-talk from jRGECO1a to the jGCaMP8m channel (= 84 peaks). (**H**) Scheme of bilateral injections of PRS×8-ChrimsonR-tdTomato into the LC of a wild-type mouse. (**I**) Epifluorescence image of a hemi-brain slice expressing PRS×8-driven ChrimsonR-tdTomato (left). tdTomato was amplified using a cross-reacting antibody against mCherry (magenta), while the LC was visualized with an anti-TH staining (black). Dashed lines indicate the fiber position (only the lesion caused by the fiber tip is visible in this slice). Transgene expression was quantified on confocal images (right). (**J**) Pupil diameter of the mouse during bilateral optogenetic activation of the LC (red bars; 633 nm, ~5 mW, 20 ms pulses at 20 Hz for 4 s). Insets show *DeepLabCut*-traced videographic images at indicated time points before and after an optogenetic stimulus. (**K**) Relative pupil diameter (color coded) during a recording session consisting of 30 trials with an inter-trial-interval of 30s. (**L**) Pupil size in response to optogenetic stimulation (mean ± standard error of the mean across trials shown in K). Absence of locomotion is shown in gray (bottom).

Finally, we developed a construct encoding the red-light-sensitive cation channel ChrimsonR [63] fused to tdTomato [64] under control of the PRS×8 promoter for optogenetic activation of LC-NE neurons. After bilateral injection of rAAV2/9-PRS×8-ChrimsonR-tdTomato (efficacy: 79.1%, specificity: 55.4%) and implantation of optical fibers above the LC in a wild-type mouse (Fig 6H and 6I), we illuminated the LC (633 nm) while monitoring the pupil (Fig 6J). As expected, illumination of the LC reliably induced pupil dilation, confirming functional optogenetic activation of the LC by PRS×8-ChrimsonR-tdTomato (Fig 6K and 6L). Importantly, pupil dilations occurred in the absence of locomotion (Fig 6L, gray), confirming direct pupil control of the LC.

## Discussion

The LC-NE system consists of a small number of neurons located deep in the brainstem. It is responsible for the modulation of a wide range of body and brain functions. In recent years, an increasing number of studies have used genetic strategies to selectively and specifically target the LC-NE system. Importantly, the interpretation of these studies depends on the reliability of transgene expression in this small cell population. However, a precise quantification of these parameters is not always reported, and a direct comparison of these model systems is lacking. In this study, we performed a side-by-side comparison of four strategies for virus-based genetic manipulation of the mouse LC, encompassing LC transduction with floxed transgenes in *Dbh^cre*, *Net^cre*, and *Th^cre* transgenic driver lines, and transgene expression under the control of a synthetic, NE-specific promoter (PRS×8).

Using immunohistological analysis, we found differences in both expression efficacy and specificity between these model systems. *Dbh^cre* mice and *Net^cre* mice showed the highest specificity and efficacy of transgene expression, with no significant differences between them. In comparison, PRS×8 promoter-driven expression showed comparable efficacy to *Dbh^cre* and *Net^cre* mice, but lower specificity compared to *Dbh^cre* mice. Conditional expression in *Th^cre* mice—a transgenic line commonly used to target LC—showed both lower efficacy and specificity than the other approaches. Finally, substantial ectopic transgene expression in regions distant to the LC was observed, particularly with PRS×8- and *Th^cre*-mediated transgene expression. Some regions, such as the parabrachial or the vestibular nucleus, frequently showed ectopic expression. While we cannot exclude a molecular predisposition of these neurons to TH- or PRS×8-mediated transgene expression, we propose a different reason: As both the parabrachial and the vestibular nucleus also frequently expressed non-conditional fluorophores (see Figs 4 or S9, especially upon injection of rAAV2/2 or low volumes of rAAV2/9), these regions might simply be better accessible for the viral suspension, likely due to the neuronal density in these regions, or due to the arrangement of nuclei and fiber tracts in the brainstem.

It is important to note that TH-positive neurons have been reported in brain areas close to the LC, such as the parabrachial nucleus [65]. Although the ectopic expression in our study was not localized to these TH-positive neurons (as indicated by the absent TH co-staining), this population should be kept in mind as it may lead to erroneous interpretations of the function of LC-NE neurons if LC-specificity is taken for granted in *Th^cre* mice. Ectopic transgene expression was much less pronounced in *Net^cre* mice, and only very low ectopic expression was observed in *Dbh^cre* mice. In fiber photometry experiments that directly target the LC, ectopic expression may be less of a concern because signals are typically recorded in close proximity to the tip of the optical fiber [66,67], adding spatial precision to the molecular specificity of transgene expression. By contrast, recordings in putative projection sites may get contaminated by afferent innervation from ectopically expressing neurons. In addition, in optogenetic studies, where the intensity of activation light is typically chosen to be well above threshold, light scattering could also lead to the activation of opsins ectopically expressed in tissue distant to the fiber tip.

As mentioned above, a direct side-by-side comparison of virus-mediated LC transduction under comparable conditions (i.e., same viral titers, injection volumes, viral serotypes, etc.) was missing. However, several different studies reported the transduction efficiency and specificity individually for a given model system. In *Dbh^cre* mice, viral transduction efficacies between 83% and 96% of TH-immunoreactive neurons have been reported [68–70], with a specificity of 95% [69]. In

*Net^cre* mice, an efficacy of ~99% was reported, with no transgene-expressing cells found outside of the LC [21]. However, it is important to note that transgene expression in this study was not achieved by viral injections but by crossing *Net^cre* mice with a reporter mouse line. With respect to PRS×8 promoter-mediated expression, specificities between 95% and 98% were reported in mice, while efficacy was not quantified [26,27]. In addition, a more recent study by Stevens and colleagues [44] reported efficacies ranging from ~50% to 90%, with specificities between ~60% and 95% (depending on the volume and titer of the injected virus). This study reported the best results with a titer of $1 \times 10^{11}$ gc/ml (71%–77% efficacy, ~96% specificity). At higher titers ($1 \times 10^{12}$ gc/ml), specificity decreased to 60%. At even higher titers ($5-20 \times 10^{12}$ gc/ml) the efficacy also decreased, and substantial toxicity was observed. The titer used in our study ($8 \times 10^{11}$ gc/ml) comes closest to the $1 \times 10^{12}$ gc/ml condition of Stevens and colleagues, and comparable efficacy (70%–90%) and specificity (60%–80%) were observed. Therefore, it is likely that the specificity of PRS×8 driven expression can be improved in the future by using lower titers. However, it is important to note that the majority of studies used PRS×8-driven constructs in combination with recombinant adenoviruses [25] or CAV-2 [26–28,44], which have a different tropism and transduction efficacy than rAAV2/9 used in our study [71]. Considering rAAV pseudotypes, insufficient PRS×8-mediated transgene expression was reported with rAAV2/7 [72] (~20.6% efficacy), whereas robust expression was reported after injection of rAAV2/9 [43,58]. These findings highlight the importance of viral serotype and careful adjustment of the viral titer when implementing the PRS×8 promoter for genetic manipulation of the LC.

In *Th^cre* mice, we show that cre-mediated expression only partially covers TH-positive LC neurons. Low efficacy and ectopic, cre-dependent transgene expression were also previously described in close proximity to the LC [22,73], the substantia nigra [22], the VTA [39,53], and various brain regions not directly related to catecholaminergic signaling [23,39,74]. A detailed study by Lammel and colleagues [39] reported specificities of 48 and 59% in the VTA of *Th^IRES-cre* and *Th^cre* mice, respectively, which aligns well with the specificity we observed in the LC. Higher efficacy and specificity for *Dbh^cre* when compared to *Th^cre* mice was also reported for catecholaminergic neurons in the A1/C1 area of the brainstem, but differences were not as pronounced as in our study [75]. Overall, our results are consistent with previous findings from various different studies investigating cre-dependent expression in the LC. Thus, our systematic comparison revealed marked differences in expression patterns of cre driver lines and promoter-mediated transgene expression of the LC-NE system.

The observed differences in transgene expression patterns may be explained, at least in part, by the different strategies used to generate different mouse lines: *Dbh^cre* mice were generated using a knock-in/knock-out strategy, resulting in cre expression directly from the *Dbh* locus. While this approach disrupts endogenous *Dbh* expression, the use of heterozygous *Dbh^cre* mice allows for safe and specific cre expression with a low risk for unwanted side effects. In contrast, the transgenic *Th^cre* mice were generated by random insertion of a construct containing the *Th* promoter followed by a cre coding sequence. This could explain leaky cre expression due to potential activity of other promoter elements in the vicinity of the integration site. However, also different strategies for *Th*-specific cre-expression yielded low specificity in the VTA—as shown for knock-in mice expressing Cre-recombinase from the downstream-untranslated region of the endogenous *Th* gene via an internal ribosomal entry sequence (IRES). The *Net^cre* mice (and other commonly used *Dbh^cre* and *Th^cre* lines [16]) are bacterial artificial chromosome (BAC) transgenic mice generated by random insertion of a BAC sequence containing the gene of interest and a cre coding sequence. The inclusion of the gene increases the possibility of preserving key regulatory elements required for accurate cell-type-specific expression, which are not included in the case of *Th^cre* mice or when using the synthetic, minimal PRS×8 promoter. However, it is important to note that regulatory elements are often distant from the promoter and therefore incomplete, and ectopic transgene expression similar to traditional transgenic mice may also be found in BAC transgenic mice [16]. Unpredictable ectopic expression may also be driven by transcriptional regulatory elements of genes flanking the transgene insertion site. In addition, random transgene insertion may interrupt endogenous genes, resulting in loss-of-function mutations and unexpected phenotypes. These effects are avoided in targeted knock-in alleles, as was used to generate the *Dbh^cre* mice. Notably,

additional *Th^cre* and *Dbh^cre* mouse lines exist that have also been generated using BACs [18] and targeted mutations [23]. However, the use of these mice is much less widespread and there are currently no vendors maintaining colonies of these lines.

Ectopic expression is not limited to the *Th* promoter, but has also been reported in a wide variety of model systems, including cholinergic and catecholaminergic cells of the retina [76,77], dopaminergic cells of the dorsal raphe nucleus [78] and the limbic system [79], and somatostatin-expressing interneurons [80] and microglia [81] in the cerebral cortex. Such ectopic expression may result from promoters that are weakly active, leading to low levels of native protein expression, which may not be functionally relevant. Similarly, low expression levels of a direct reporter, such as a fluorophore in a reporter mouse line, may not be a major concern because the detected signal may be much weaker compared to true positive cells exhibiting high activity of the promoter of interest. However, in cre-reporter lines, even low levels of cre are sufficient to recombine floxed transgenes over time, leading to substantial expression of the transgene and thus uncoupling of endogenous cre expression levels from cre-dependent transgene expression [82,83]. This explanation is consistent with reports from homozygous Ai32 reporter mice where immunostainings revealed leaky channelrhodopsin-2 expression throughout the brain in the absence of cre, but at levels insufficient to induce detectable photocurrents [84]. Other explanations for ectopic expression include legitimate but transient promoter expression during development [23,30,31,39,83], changes in endogenous expression patterns resulting from transgene integration itself [76], or recombination of cre in the germline [83].

Phenotypic alterations have been previously reported for different cre driver lines to access the cholinergic (*Chat^cre*) [46] and dopaminergic systems (*Dat^cre*, *Dat^IREScre*, and *Th^cre* upon amphetamine application) [47,48]. Therefore, we performed a behavioral screening sensitive to anxiety-like behaviors as well as learning and memory, all of which have been linked to LC function. We found no genotype-related effect on spontaneous locomotion, anxiety, exploratory behavior, working memory, and spatial learning and memory. These results are particularly relevant for the *Dbh^cre* line, because the knock-in/knock-out design of this mouse line disrupts endogenous *Dbh* expression in one allele of heterozygous animals. This approach could lead to reduced levels of NE while accumulating its precursor molecule DA, and consequently disrupt behavior. Furthermore, we did not observe any sex-related effects on anxiety-like behavior, which is in contrast to a recent study reporting higher levels of anxiety in female mice using the open field and EPM paradigms [85]. While sex differences in the EPM were confirmed by one additional study [86], two studies reported the absence of sex effects on anxiety-like behavior using the open field test [86,87]. These discrepancies may be explained (in part) by the estrous cycle, which is linked to *Th* transcription [88]. It has been demonstrated that female mice in the proestrus and estrus stages showed increased anxiogenic behavior, while female mice in other stages were comparable to males [85]. As the stage of the estrous cycle was not controlled in our experiments, experimental groups randomly composed of females in different estrous cycles may have masked sex-dependent effects in our experiments. Further factors that have been shown to influence the outcome of these experimental paradigms are the sex of the experimenter [89,90], the history of handling [91,92], as well as non-explicit factors of the laboratory itself [93,94]. In conclusion, our results suggest that all mouse lines investigated here can be used equally, at least for the behavioral assays we have included. However, caution should be exercised when extrapolating these results to other experimental paradigms such as instructed behavior or experimental conditions that differ substantially from those we used.

We show that *Dbh^cre*, *Net^cre*, and PRS×8-mediated transgene expression provides reliable access to the LC-NE system. In parallel, we did not observe any genotype-related behavioral changes in any mouse line. Apart from these aspects, additional factors should be considered when deciding which LC-targeting strategy to use. First of all, *Dbh^cre* and *Net^cre* mice showed the highest specificity. Second, the cre-lox system is widely established. Therefore, cre driver lines can be easily combined with off-the-shelf viral constructs that are readily available from public commercial and non-commercial repositories. Finally, driver lines can be crossed with various reporter lines, allowing reproducible transgene expression without the need for stereotactic surgery. Of note, *Dbh^cre* mice are commercially available in public repositories, making

them available to the scientific community, while a material transfer agreement is required to obtain *Net^cre* mice. In contrast, PRS×8-based transgene expression is possible in wild-type mice and, therefore, no transgenic mouse line colonies are required. This not only eliminates the risk of mouse-line-dependent phenotypic effects that we may have missed, but also facilitates easy and straightforward access to the LC-NE system. Moreover, it reduces the number of excess animals that can result from unwanted genotypes when breeding a particular mouse line. In addition, PRS×8 allows a high degree of flexibility for manipulation and readout of different molecularly defined circuits when combined with cre lines other than those related to the noradrenergic system [32]. Finally, the PRS×8-system can be implemented in species other than the mouse, as recently demonstrated in the rat [28].

Care should be taken when translating our findings to other model systems, such as rats, where other approaches than those reported in this study are also available. For example, transgene expression mediated by the PRS×8 promoter resulted in an efficacy of 83% of TH⁺ neurons [28] at a specificity of 97%–100% in this model system [25,28,58]. While this efficacy is in line with our findings, the reported specificities exceed the one of our experiments, which might be due to the injected titers (as discussed above) [44], or the (sero)type of the virus used, as many studies in rats relied on CAV-2 instead of rAAVs [28,44]. Furthermore, *Th^cre* transgenic rats have been created over a decade ago [24], and a specificity of 97%–100% was reported for viral transduction of the LC-NE system, while efficacy amounted to 73%–75% [24,95]. Most recently, a novel *Dbh^cre* rat line has been created and characterized with both high specificity (97%–99%) and efficacy (95%–98%) [20], thereby offering one more approach to target the LC-NE system in rats with high precision. However, regardless of the model system chosen, our study highlights the importance of rigorous post-mortem verification of viral transduction and the implementation of appropriate controls for each system.

It is also important to consider viral titers and pseudotypes when extrapolating our conclusions to different experimental settings. First, we relied on a single viral titer in all model systems. While all experimental groups were injected with the same amount of virus, it has been shown that efficacy, specificity, and potential toxicity of PRS×8-mediated gene expression are dependent on the viral titer [44]. Similarly, the specificity of cre-dependent transgene expression is dependent on the viral titer [96,97]. Therefore, it may well be that greater efficacy and specificity of gene expression could be achieved when optimizing the viral titer for a specific application. In addition, the majority of our experiments were performed with rAAV2/9, and, therefore, may not be generalizable to other serotypes or even different virus families. For example, an rAAV pseudotype with less broad tropism than rAAV2/9 (e.g., rAAV2/2) [42] may lead to reduced ectopic expression and, therefore, higher apparent specificity. Conversely, lower efficacy may be observed, if viral transduction of noradrenergic neurons is limited due to low tropism of the virus.

Another consideration relates to the choice of antibodies for the detection of transgene expression and cell identity. For example, we have broadly stained against TH while anti-DBH or anti-NET antibodies could have been used to exclusively detect noradrenergic neurons. However, all of our experimental groups were stained with the same antibody, and the outcome of our study did not favor the *Th^cre* line.

In conclusion, we present a quantitative side-by-side comparison of four different strategies to genetically access the mouse LC, resulting in heterogeneous patterns of transgene expression. These findings may assist the design of future studies and the interpretation of results obtained from previous studies using any of these model systems. For example, a recent study described micro-arousals, but no sustained wake episodes upon optogenetic LC activation in *Dbh^cre* mice, whereas sustained wake episodes were evoked by optogenetic LC activation in *Th^cre* mice [73]. One possibility is that this discrepancy could be the consequence of different neuronal populations targeted in each line, and highlights the importance of selecting an appropriate model system. In addition to the analysis of transgene expression patterns, we developed AAV-based tools relying on the PRS×8 promoter for marking, monitoring, and manipulating LC-NE neurons, thereby increasing experimental flexibility and facilitating the investigation of the LC-NE system in various genetic backgrounds. Taken together, our results contribute to the design of more refined studies and thus to a deeper understanding of the heterogeneous functionality of the LC-NE system in the future.

## Materials and methods

### Ethics statement

All experiments were performed in compliance with German law according to the directive 2010/63/EU of the *European Parliament* on the protection of animals used for scientific purposes and in accordance with the 3R principles. Procedures were approved by local animal care committees (*Behörde für Gesundheit und Verbraucherschutz*, Hamburg, Germany: N033/2019; *Regierungspräsidium Karlsruhe*, Germany: 35-9185.81/G-213/22).

### Animals

Experiments were performed on adult (2–6 months of age) C57BL/6J, *Th^cre* [22], *Net^cre* [21], and *Dbh^cre* [19] mice, housed in cages with nesting material with no more than five animals per cage at the animal facility of the *Center for Molecular Neurobiology* of the *University Medical Center Hamburg-Eppendorf* or at the *Medical Faculty Mannheim of the University of Heidelberg*. Animals were kept in a 12/12 h light/dark cycle, at 20–24 °C and 40%–65% humidity, with access to water and food ad libitum.

### Molecular biology

The DIO variant for cre dependent expression of eGFP [98] was created by cloning eGFP in antisense direction, flanked by two loxP and lox2272 sites, behind the CAG promoter into an AAV2 backbone. PRS×8-based constructs were created by cloning the respective transgene behind 8 repetitions of a de novo synthetized Phox2a/Phox2b response site (PRS; based on the only PRS×8 construct (#89539) available from *AddGene* at the time this study started) into an AAV2 backbone. PRS×8 constructs were generated for the expression of the green light emitting reporter protein eGFP [98] (pAAV-PRS×8-eGFP; *AddGene* #192589), the green light emitting calcium indicator jGCaMP8m [59] (pAAV-PRS×8-jGCaMP8m; *AddGene* #210400), and the red-light sensitive cation channel ChrimsonR coupled to the red light emitting reporter protein tdTomato [63] (pAAV-PRS×8-ChrimsonR-tdTomato; *AddGene* #192587). DNA was then packed into recombinant adeno-associated virus serotype 2/9 (rAAV2/9) at the vector core facility of the *University Medical Center Hamburg-Eppendorf*, Germany, or by *VectorBuilder*.

### Stereotactic injections and fiber implantation

For surgical procedures, mice were either anesthetized with Midazolam/Medetomidine/Fentanyl (MMF; 5/0.5/0.05 mg/kg body weight in NaCl, intraperitoneal), or with Isoflurane (5% for induction, ~1.5% for maintenance). In the case of isoflurane anesthesia, adequate analgesia was achieved with buprenorphine (0.1 mg/kg in NaCl, intraperitoneal). After confirming anesthesia and analgesia by the absence of the paw withdrawal reflex, the head was shaved with a fine trimmer and sterilized with iodine (*MundiPharma*). Mice were then fixed with an auxiliary ear bar (EB-6, *Narishige Scientific*) in a stereotactic frame and positioned on a heating blanket to maintain body temperature. Eyes were covered with an eye ointment (*Vidisic*, *Bausch + Lomb*) to avoid drying. Subsequently, the scalp was opened (~1 cm incision) and the skull was cleaned with NaCl (0.9%; *Braun*) and a bone scraper (*Fine Science Tools*). Bregma and lambda were identified and stereotactically aligned. A small craniotomy was then performed above the LC (AP: −5.4; ML: ± 0.9; DV: −3.6 mm relative to bregma; for later injections, the mediolateral coordinates were changed to ±1.1 mm, see Fig 4) using a dental drill (*Foredom*), and was constantly kept moist to prevent drying.

The viral suspension was drawn into a borosilicate glass pipette (pulled from 0.2 mm glass capillaries with a PC-100 puller; *Narishige*) using negative air pressure. 300 nl of titer-matched ($8 \times 10^{11}$ gc/ml) viral suspension containing rAAV2/9 carrying DNA encoding *eGFP* either in a cre-dependent (DIO) manner under control of the CAG promoter (in the case of *Dbh^cre*, *Net^cre*, and *Th^cre* animals) or unconditional under control of the PRS×8 promoter (in the case of C57BL/6J mice) were bilaterally injected into the LC with an injection speed of ~100 nl/min using a custom-made, air-pressure driven

injection system for the analysis of transgene expression in the different model systems. For viral transduction of the VTA, 200 nl of virus suspension were injected at four different injection sites (AP: −3.2; ML: ±0.3; DV: −3.5 mm and AP: −3.2; ML: ±0.5; DV: −3.2 mm relative to Bregma). For investigations of promoter effects, 300 nl of rAAV2/9 carrying DNA encoding eGFP in a cre-dependent manner under control of the hSyn promoter were injected in $Dbh^{cre}$ mice at a titer of ($8.2 \times 10^{12}$ gc/ml). For investigating the effects of viral serotype and injection volume on viral spread, 300 nl of rAAV2/2 and 300/100/50 nl of rAAV2/9 carrying DNA encoding eGFP under control of the hSyn promoter were injected in wild-type mice in a titer-matched manner at $6.8 \times 10^{12}$ gc/ml. Injections of rAAV2/9-Ef1a-DO-DIO-tdTomato-eGFP ($2.3 \times 10^{13}$ gc/ml; *AddGene* plasmid #37120) [45] into the LC were performed in $Dbh^{cre}$, $Net^{cre}$, and $Th^{cre}$ animals. For fiber photometry experiments, we injected rAAV2/9-PRS×8-jGCaMP8m ($3 \times 10^{12}$ gc/ml) and rAAV2/9-hSyn-DIO-jRGeco1a ($2 \times 10^{13}$ gc/ml) into the LC of a $Dbh^{cre}$ mouse. For optogenetic manipulation of the LC, 300 nl of rAAV2/9-PRS×8-ChrimsonR-tdTomato ($1 \times 10^{12}$ gc/ml) were injected into each LC of a wild-type mouse. After virus injection, the glass micropipette was kept in place for at least 5 min to avoid unnecessary viral spread and was then slowly withdrawn. In the case of histological experiments, the skin was closed with simple interrupted sutures and covered with iodine. In the case of fiber photometry experiments, optical fibers (R-FOC-BL400C-50NA: 400 µm diameter, 0.5NA; *RWD*) were inserted at AP: −5.4; ML: ±1.0; DV: −3.5 mm relative to bregma using a stereotactic micromanipulator. For optogenetic experiments, optical fibers (R-FOC-BL400C-37NA: 200 µm diameter, 0.37 NA; *RWD*) were inserted at the same coordinates. Fiber implants and a metal headpost (00-200 500 2110; *Luigs & Neumann*) for animal fixation during the experiment were fixed to the roughened skull using cyanoacrylate glue (Pattex; Henkel) and dental cement (Super Bond C&B; *Sun Medical*). The incised skin was then glued to the dental cement to close the site, before it was covered with iodine. Mice received analgesia (Carprofen; 4 mg/kg body weight, subcutaneous), and Atipamezole/Flumazenil/Buprenorphine (FAB) was administered (2.5/0.5/0.1 mg/kg body weight, intraperitoneal) to antagonize anesthesia in the case of MMF; Isoflurane anesthetized mice were simply removed from the Isoflurane. For recovery, mice were placed in a cage which was placed on a heating pad. Meloxicam was mixed with soft food for the following two days.

## Tissue collection

For perfusions, mice were deeply anesthetized with Ketamine/Xylazine (180/24 mg/kg body weight, intraperitoneal), which was confirmed by the absence of the paw withdrawal reflex. Mice were fixed on a silicone pad and transcardially perfused with ~50 ml of phosphate-buffered saline (PBS) to rinse out the blood and subsequently with ~50 ml of 4% paraformaldehyde (PFA; pH 7; *Carl Roth GmbH*) to fix the tissue. Brains were explanted and post-fixed in 4% PFA for at least 24 h. After fixation brains were embedded in 3% agarose (*Genaxxon Bioscience*) and coronal slices of ~50 µm thickness were obtained using a vibratome (VT 1000S, *Leica*). Slices were stored in PBS at 4 °C until they were stained and mounted on microscope slides.

## Immunohistochemistry

To block unspecific binding sites for antibodies, brain slices were incubated in 10% normal goat serum (NGS) and 0.3%–0.5% Triton X-100 in PBS for 2 h at room temperature. Slices were then incubated for 2 days at 4 °C in a carrier solution (2% NGS, 0.3%–0.5% Triton X-100, in PBS) containing primary antibodies (see below) to allow for sufficient target protein-primary antibody interaction. Subsequently, slices were rinsed three times for 5–10 min in PBS to remove unbound primary antibodies. Following the washing steps, slices were incubated with secondary antibodies (see below) in the above-mentioned carrier solution over night at 4 °C. Washing steps were repeated as described before in order to remove unbound secondary antibodies. Finally, slices were mounted on microscope slides using *Fluoromount* mounting medium (*Serva*, Germany).

To analyze eGFP expression in TH+ cells of the LC (Figs 1, 2, 3A–3D, S1, S4, S6, and S8), slices were incubated with primary antibodies against TH (1:1000; AB152; *Invitrogen*, *Thermo Fisher Scientific*) derived from rabbit and against

green fluorescent protein (1:750; A10262; *Invitrogen*, *Thermo Fisher Scientific*) derived from chicken, while Alexa Fluor 546 goat anti-rabbit (1:1000; A11035; *Invitrogen*, *Thermo Fisher Scientific*) and Alexa Fluor 488 goat anti-chicken (1:1000; A11039; *Invitrogen*, *Thermo Fisher Scientific*) secondary antibodies were used. To analyze the difference between natively expressed eGFP fluorescence and antibody-stained eGFP fluorescence (S7 Fig), slices were incubated with a primary antibody against green fluorescent protein (1:750; A10262; *Invitrogen*, *Thermo Fisher Scientific*) derived from chicken and an Alexa Fluor 546 goat anti-chicken secondary antibody (1:1000; A11040; *Invitrogen, Thermo Fisher Scientific*). In DO-DIO-experiments (Fig 4), slices were incubated with primary antibodies against TH (1:1000; AB152; *Invitrogen*, *Thermo Fisher Scientific*) derived from rabbit and against green fluorescent protein (1:750; A10262; *Invitrogen*, *Thermo Fisher Scientific*) derived from chicken, while Alexa Fluor 647 goat anti-rabbit (1:1000; A32733; *Invitrogen*, *Thermo Fisher Scientific*) and Alexa Fluor 488 goat anti-chicken (1:1000; A11039; *Invitrogen*, *Thermo Fisher Scientific*) secondary antibodies were used. No staining was performed against tdTomato. To visualize jGCaMP8m in TH$^+$ neurons of the LC (Fig 6B), slices were incubated with primary antibodies against TH (1:1000; AB152; *Invitrogen*, *Thermo Fisher Scientific*) derived from rabbit and against green fluorescent protein (1:500; A10262; *Invitrogen*, *Thermo Fisher Scientific*) derived from chicken, while Alexa Fluor 647 goat anti-rabbit (1:1000; A32733; *Invitrogen*, *Thermo Fisher Scientific*) and Alexa 488 goat anti-chicken (1:1000; A11039; *Invitrogen*, *Thermo Fisher Scientific*) were used as secondary antibodies. To visualize jRGeco1a in TH$^+$ neurons of the LC (Fig 6B), slices were incubated with primary antibodies against TH (1:500; Cat. No. 213 104; *Synaptic Systems*) derived from guinea pig and against DsRed (1:500; Cat. No. 632496; *TaKaRa Bio*) derived from rabbit, while Alexa Fluor 647 goat anti-guinea pig (1:200; A21450; *Invitrogen*, *Thermo Fisher Scientific*) and Alexa 546 goat anti-rabbit (1:1000; A11035; *Invitrogen*, *Thermo Fisher Scientific*) were used as secondary antibodies. To visualize ChrimsonR-tdTomato in TH$^+$ neurons of the LC (Fig 6I), slices were incubated with primary antibodies against TH (1:1000; AB152; *Invitrogen*, *Thermo Fisher Scientific*) derived from rabbit and against mCherry (1:500; Cat. No. 632543; *TaKaRa Bio*) derived from mouse, while Alexa Fluor 647 goat anti-rabbit (1:1000; A32733; *Invitrogen*, *Thermo Fisher Scientific*) and Alexa Fluor 488 goat anti-mouse (1:1000; A32723; *Invitrogen*, *Thermo Fisher Scientific*) were used as secondary antibodies.

## Image acquisition

Confocal images of the LC were acquired in a Z-stack (2 μm step size) using an LSM 900 airyscan 2 microscope (*Zeiss*, Germany), controlled by the *ZEN* 3.1 imaging software, with laser intensities and gain of the photomultiplier tubes adjusted to maximize dynamic range while avoiding saturation of images. A Plan-APOCHROMAT 20X/0.8 objective was used, resulting in a field of view of 319.45 × 319.45 μm with a resolution of at least 1,744 × 1,744 pixels (183.2 nm/pixel). Some images were taken at a resolution of 2,044 × 2,044 pixels (i.e., 0.156 nm/pixel), and subsequently down-sampled to match the resolution of 1,744 × 1,744 pixels. Whole slices were either imaged using a *Zeiss AxioObserver* epifluorescence microscope (*Zeiss*, Germany) with a Plan-APOCHROMAT 20×/0.8 objective, controlled by the *ZEN* 3.3 imaging software (ectopic transgene expression: Figs 2 and S8, verification of LC transduction: Fig 4, leakage of viral transduction: S4 Fig) or using a *Zeiss Axio Scan Z.1* epifluorescence microscope (*Zeiss*, Germany) with a Plan-APOCHROMAT 20×/0.8 objective, controlled by the ZEN 3.1 imaging software (viral spread: Figs 3 and S9). *ImageJ* [99] was used to adjust image brightness, contrast, and create maximum intensity Z-projections.

## Image analysis

*Transgene expression* was detected at the cellular level using the deep-learning-based algorithm *CellPose* [35]. In the first run, we analyzed all images from all approaches semi-automatically (i.e., automatically segmented cells were manually curated). We then used the obtained cell masks to re-train *CellPose* [36] in order to adapt it to our dataset. We have uploaded this custom-trained CellPose model, named "CP4LC_Wissing_Eschholz_et_al" as S1 Material. In the second run, we used the re-trained model for automatic cell segmentation without any manual annotation. Segmentation

was done separately for each channel, resulting in a total of 2766/3638/2774/2699 TH$^+$ cells and 2293/4195/1861/3061 eGFP$^+$ cells in *Dbh$^{cre}$*, *Net$^{cre}$, Th$^{cre}$*, and PRS×8-animals, respectively (33/39/31/30 brain slices; 11/12/11/10 transduced LC; $n = 7/7/7/7$ mice). Please note that in our study transgene expression of calcium indicators and optogenetic tools were harder to quantify in an automated way, as transgene expression was not as bright and hence the delineation of individual cells not as clear as compared to the expression of eGFP. The average number of TH$^+$ cells that were found per slice was 83.8 ± 19.3, 93.3 ± 15.7, 89.5 ± 16.0, and 90.0 ± 15.3 cells (mean ± standard deviation) for *Dbh$^{cre}$*, *Net$^{cre}$, Th$^{cre}$*, and PRS×8, respectively. Cell detection was cross-validated with quantification by two different experimenters, who counted stained cells with the cell counter function of *ImageJ*. Cell masks of segmented cells in the *eGFP* and *TH* channels exported from *CellPose* as.txt-files were then checked for overlap using custom-written *MATLAB* scripts (*The MathWorks*; see data availability statement). For quantification of transduction efficacy, co-expression was checked based on the cell masks in the *TH* channel, and masks which had an overlap of ≥50% with the masks in the *GFP* channel were defined as co-expressing. Similarly, co-expression was quantified based on the cell masks in the *GFP* channel, in order to quantify the specificity of transduction. The number of TH$^+$, GFP$^+$, and TH-GFP-co-expressing cells was subsequently summed up per animal, in order to avoid biasing the results based on (i) the different amount of TH$^+$ cells on individual slices and (ii) the total amount of slices taken from each animal (some slices had to be excluded due to tissue damage during the mounting process). Quantification of the effect of anti-GFP immunostaining on cell fluorescence was performed by fitting (least-squares) a binormal model of the form $f_R = \phi(\text{bias} + \text{gain} \times \phi^{-1}(f_G)) + \varepsilon$ with $\varepsilon \sim N(0,\sigma^2)$; and where $f_R$ and $f_G$ denotes the fluorescence in the red (GFP-stained) and green (native eGFP) channels respectively, and $\phi$ corresponds to the normal cumulative distribution function. Comparisons between the two channels have been performed based on the cell masks detected in the GFP-stained channel, as we aimed to reveal additional cells upon GFP staining, which, by definition, are not detected in the native eGFP channel. To quantify v*iral spread*, images were down-sampled to 25% of their original size using bicubic interpolation (MATLABs built-in function '*imresize*') and smoothed using a 2D gaussian filter with a kernel width of 50 μm ('*imgaussfilt*') before fluorescence was normalized from 0 to 1. Afterward, viral spread was defined as cumulative areas of the image in which fluorescence exceeded a certain threshold, while thresholds were varied between 0 and 1 at an increment of 0.1.

## Behavioral screening

All behavioral experiments were performed in a quiet, darkened room by the same (female) experimenter, blind to the animal's genotype. Animals (genotype-, age- and sex-matched littermates, max. 2 animals of the same genotype and sex per litter) between 12 and 15 weeks of age at the start of the experiment were handled for 5 min (manually picked up and let to freely explore the experimenter's hands and arms) every other day in order to habituate mice to the experimenter. All experiments were video-recorded with a regular (when experiments were performed under illumination) or an infrared camera (when experiments were performed in darkness). All reported measures were performed using the software *Etho-vision* XT17 (*Noldus*, the Netherlands), unless stated otherwise. *Open Field Test.* An open field arena of 50 × 50 × 40 cm, homogeneously illuminated with white light of 25 lux, was used. At the start of the experiment, mice were placed in one corner of the box, and their movement was tracked for the following 20 min. The time spent in the center (the central 20 × 20 cm) and borders (5 cm from the walls) of the arena, the number of center crossings, and the latency to enter the arena center for the first time served as proxies for anxiety-like behavior, while the total travel distance served as a proxy for spontaneous locomotion. *Elevated Plus Maze*. The EPM consisted of four arms (30 × 5 cm) radiating from a central platform (5 × 5 cm), elevated 75 cm from the ground. Two opposing arms are covered by opaque walls (height: 17 cm), while the two other arms remain open. Mice were placed on the central platform and left to explore the maze for 5 min in total darkness. The time spent on opened versus closed arms, the number of entries to open arms, the fraction of open arm entries upon which mice went to the distant edge of the exposed arm, and the latency to leave the center of the maze served as proxies for anxiety-like behavior. In addition, the number of rearing events, the latency to the first rearing event,

and the percentage of time spent grooming served as indirect measure for the absence of anxiety. EPM data acquisition and analysis were performed using *Observer* XT11 (*Noldus*, the Netherlands). *Y-maze*. The maze consisted of three symmetrically radiating arms (34 × 5 cm), surrounded by walls (height: 30 cm) made of transparent plastic, with spatial cues placed behind each wall, homogeneously illuminated with white light of 10 lux. Mice were placed in the center of the maze and left to explore the maze until 26 transitions (entry to one arm with all four paws) between neighboring arms were made, or for a maximum of 15 min. Entering an arm after having visited the two other arms before, that is entering all three arms of the maze in one sequence, was considered a complete alternation. The fraction of alternations over all transitions (a proxy for working memory) as well as the time per transition (a proxy for exploratory behavior) was manually extracted. *Morris Water Maze*. The water maze consisted of a circular pool (diameter: 145 cm) filled with opaque water (19–21 °C) with non-toxic white paint, surrounded by black curtains, and homogeneously illuminated with white light of 60 lux. A platform (diameter: 14 cm) was placed 1 cm below the water surface in one of the four quadrants of the maze. Four landmarks (35 × 35 cm) of different shapes and gray tones were placed on the walls of the maze to allow for orientation. Animals were familiarized with swimming, platform searching, and handling in a pre-training session using a small, rectangular water tank. Mice had to search for a platform in three trials (10–20 min inter-trial intervals), with a different platform location each time. Animals were transferred back to their home cage, where heat lamps were provided. On training days 1 and 2, animals underwent four training trials each (max. 90s), with varying starting positions. Once they climbed the platform, they were left to wait there for 10 s before being transferred back to the home cage. Animals that did not find the platform within 90 s were guided to the platform until they climbed it and left to wait on the platform for 10 s, before being transferred back to the home cage. On day 2, after the second training trial, and on day 3, a probe trial (60 s duration) without the platform was conducted. The time spent in the area where the platform was previously located, the time spent in the quadrant that previously contained the platform, the latency to the platform area, and the average minimum distance to the platform area were used to approximate successful spatial learning and memory. The total travel distance served as a measure for locomotor behavior, and the average distance to the wall was analyzed to compare thigmotactic behavior as well as searching strategies (i.e., circling around the pool).

## Fiber photometry recordings

Approximately 3 weeks after surgery, animals were habituated to the experimenter, set up, and head fixation on a linear treadmill (200–100 500 2100 and 700–100 100 0010; *Luigs & Neumann*). Fiber photometry recordings were performed using two *pyPhotometry* systems [100] (Open *Ephys*) controlling multicolor LED light sources (pE-4000; *CoolLED*) connected to the input ports of dichroic cubes (FMC5_E1(400–480)_F1(500–540)_E2(555–570)_F2(580–680)_S; *Doric Lenses*) as reported previously [13]. In brief, excitation light for quasi-isosbestic (405/10 nm) and calcium-dependent (470/10 nm) activation of jGCaMP8m was delivered in a temporally interleaved manner into the 400–480 nm input port of the dichroic cube, while excitation light for jRGECO1a (555–570 nm) was delivered via the 555–570 nm port. We also included excitation at 405/10 nm as a control channel when using jRGECO1a, as the indicator has been shown to be largely insensitive to calcium when excited at this wavelength [101], but we note that the efficiency of jRGECO1a excitation at 405 nm is very low (which, to a milder degree, is also true for jGCaMP8m). Hence, the recorded signal, which besides indicator fluorescence also contains autofluorescence originating from fiber optics and brain tissue [102], is likely composed differently and should not be over-interpreted. Emission light between 500–540 nm (jGCaMP8m) and 580–680 nm (jRGECO1a) was measured using photodetectors (DFD_FOA_FC; *Doric Lenses*), and digitized by the *pyPhotometry*-board at 130 Hz. Mice were connected to the dichroic cubes with low-autofluorescence patch cords (MFP_400/430/3000-0.57_1m_FC-ZF1.25(F)_LAF: 400 µm diameter, 0.57 NA; *Doric lenses*) via zirconia mating sleeves (SLEEVE_ZR_1.25; *Doric Lenses*). Simultaneously, the left eye of the mouse was monitored using a monochrome camera equipped with a macro-objective (DMK 22UX249 and TMN 1.0/50; *The Imaging Source*) and a 780 nm long-pass filter (FGL780; *Thorlabs*) under infrared illumination. Single video frames were triggered at 30 Hz with a voltage signal provided

by an NI-DAQ-card (PCIe-6323; *National Instruments*), which was in addition fed to the *pyPhotometry* system for data synchronization.

## Fiber photometry analysis

Single-sample outliers were removed from raw traces by applying a moving median filter with a window size of three sampling points, before traces of quasi-isosbestic and ligand-dependent fluorescence were detrended by fitting a double-exponential function to the respective trace and dividing the trace by the fitted curve, (accounting for bleaching components resulting both from the indicator and from the fiber optics [103]). Traces were then band-pass filtered (1/60–30 Hz; third order Butterworth filter) using *MATLAB*'s 'butter', and 'filtfilt' functions in order to remove signal components resulting from remaining bleaching and high frequency noise. The difference between the corrected functional and quasi-isosbestic traces was then taken as the functional signal, ΔF. Pupil edges (horizontal, vertical, and both diagonals) were extracted from each frame using *DeepLabCut* [104], and the pupil diameter was defined as the average distance of opposing edges in each frame. The pupil diameter was then smoothed with a window size of 250 ms (using the *MATLAB* functions 'resample' and 'smooth'). Animal position was recorded using the treadmill-software provided by *Luigs and Neumann*, smoothed by using a moving median filter with a window size of 500 ms, and speed was calculated as the derivative of animal position. For the experiments reported in this manuscript, the treadmill was largely immobilized to minimize locomotion-related pupil dilation. Yet, attempts of the animals to start locomotion could be detected, and trials in which treadmill movement exceeded 5 mm/s were excluded from further analysis. All data was re-sampled to 100 Hz for further processing. To detect periods of pupil dilation, peaks were detected in the derivative of pupil diameter using using *MATLAB*'s 'findpeaks' function, with a minimum peak prominence of the mean plus three standard deviations of the whole trace, and a minimum peak distance of 3 seconds. Photometry, pupil, and treadmill recordings were then cropped and aligned in a time window starting 2 seconds before and ending 5 seconds after each peak.

## Optogenetic manipulation of LC activity

General procedures for optogenetic experiments were identical to fiber photometry experiments, except for the delivery of optogenetic stimuli. Stimuli were generated using custom-written *MATLAB* scripts actuating on an NI-DAQ-card (USB-6001; *National Instruments*), controlling a 633 nm diode laser (633-100; *Omicron Laserage*) housed in a laser combiner system (LightHUB; *Omicron Laserage*). The laser combiner was coupled to the optical fiber implants using a 1 × 2 step-index multimode fiber optic coupler (200 µm core diameter, 0.37 NA; SBP(2)_200/220/900-0.37_2m_FCM-2xZF1.25; *Doric Lenses*) via zirconia mating sleeves (SLEEVE_ZR_1.25; *Doric Lenses*) wrapped with black shrinking tubes to avoid light emission from the coupling interface. To activate ChrimsonR, pulse trains (633 nm, ~5 mW at each fiber end, 20 ms pulse duration, 20 Hz repetition rate) of 4 s duration were used. Thirty stimuli were presented at an inter-stimulus-interval of 30 s. Pupil diameter was extracted from videographic images as for fiber photometry experiments, cropped, and aligned to optogenetic stimulation in a time window starting 5 s before and ending 20 s after each stimulus onset. Pupil diameter was then normalized to the median pupil diameter during the second before stimulus onset.

## Statistics

For histological data, one-way ANOVA was performed; when significant, post-hoc comparisons between groups were performed using Tukey's honestly significant difference criterion which controls for inflated risk of type-1 error (false positives) following multiple comparisons (using *MATLAB*'s 'anova1' and 'multcompare' functions). For behavioral data, two-way ANOVA was performed to identify differences between genotypes, sexes, and interactions thereof, using the software *Graphpad Prism*. All statistical tests were two-tailed, with a type-1 error risk set to $a = 0.05$. In all figures, error bars correspond to standard error of the mean, unless specified otherwise.

## Supporting information

**S1 Data. Numerical Data underlying the reported findings.**
(XLSX)

**S1 Material. Custom-trained model for *CellPose*, optimized for the segmentation of coronal LC sections of the mouse locus coeruleus virally transduced to express fluorescent proteins.**
(ZIP)

**S1 Table. Cre driver lines to target the NE system in mice.** Bold font indicates mouse lines used in this study. Gray font indicates mouse lines which are not readily available from any vendor.
(DOCX)

**S2 Table. Statistical analysis of behavioral screening.** The table indicates results of a two-way ANOVA analyzing effects of genotype (GT), sex, or interactions thereof (×) in the mouse lines included in this study on different aspects of behavior in the in the open field test (OF), elevated plus maze (EPM), Y-maze (YM), and Morris water maze (MWM). All groups included eight male and eight female mice of each genotype. *p*-values below 0.05 are indicated in bold font. Note that multiple comparisons were not corrected in this table.
(DOCX)

**S1 Fig. Additional examples of eGFP expression in *Th^cre^* mice.** TH-expression (magenta) and eGFP expression (green) in exemplary *Th^cre^* mice with high (**a**) and low efficacy (**b**). Scale bar: 50 μm.
(TIF)

**S2 Fig. Decoupling of TH and eGFP expression in the LC region of *Th^cre^* mice.** Normalized fluorescence in the green channel of neurons co-expressing TH and eGFP (i.e., true positives; left) and neurons expressing eGFP only (i.e., false positives; right) in *Dbh^cre^* (red), *Net^cre^* (magenta), *Th^cre^* (orange) and PRS×8 mice (blue). While normalized fluorescence of false positive neurons was lower as compared to true positive neurons in *Dbh^cre^*, *Net^cre^* and PRS×8 mice, normalized fluorescence of false positive neurons exceeded true positive neurons in *Th^cre^* mice. ****: $p < 0.0001$ (Bonferroni-corrected Wilcoxon rank sum test). Numerical data underlying this figure can be found in S1 Data.
(TIF)

**S3 Fig. Independence of histology quantification on the overlap threshold of cell masks.** (**a**) Efficacy and specificity from the four different model systems as a function of the threshold overlap parameter (light to dark colors; ranging from 5% to 95%, in steps of 5%) which defines the minimal overlap between GFP⁺ and TH⁺ channels for a TH⁺ cell to be also labeled as GFP⁺. (**b**) Transduction efficiency (summarizing sensitivity and specificity in a single metric) from the four different model systems quantitatively varied as a function of the threshold overlap parameter. Importantly, conclusions were robust to this parameter choice as shown by the non-crossing of the curves. In the main text, we report values for a neutral threshold overlap parameter of 50%. Shading areas denote the standard error of the mean ($n = 7$).
(TIF)

**S4 Fig. Leakage of cre-dependent virus in wild-type mice.** Immunohistochemical staining against GFP (green) and TH (magenta) in two wild-type mice injected with rAAV2/9-DIO-eGFP. eGFP expression was only observed in very few cells both within and outside of the LC.
(TIF)

**S5 Fig. Agreement between *CellPose*-based and human-based cell segmentation.** (**a**) Correlation of cell counts; (**b**) Efficacy and specificity; (**c**) transduction efficiency between cells identified by CellPose vs. labeled by each of the

two human observers (AD and CW; first 2 columns); and between the two human observers (third column). A total of five slices of five different mice were used for each of the four model systems. Dashed lines indicate identity. In each case, two measures of agreement are reported: Pearson's correlation coefficient, denoted as $r$; and the distance between observers, denoted as $d = \sqrt{\sum_i (x_i - y_i)^2}$ where $x$ and $y$ correspond to observers' reports (**a**) or derived quantities (**b**, **c**), and $i$ to the brain slice. Error bars denote the standard error of the mean (with $n = 5$).
(TIF)

**S6 Fig. Specificity of cre-dependent transgene expression in the VTA of *Th^cre^* mice.** (**a**) Injection scheme: CAG-DIO-eGFP was bilaterally injected into the VTA of *Th^cre^* mice. (**b**) Example of a coronal brain slice stained against TH (magenta) to visualize dopaminergic neurons of the VTA and eGFP (green) to enhance the native fluorescence of eGFP expression. (**c**) Exemplary confocal images of VTA-DA neurons in *Th^cre^* mice (top), along with corresponding cell masks as segmented by *CellPose* (bottom). Overlaid cell masks of both channels are shown at the bottom right, where each TH$^+$ cell was labeled as GFP$^+$ (successful transduction, black) when cell masks overlapped 50% or more the size of the TH$^+$ cell, or GFP$^-$ otherwise (missed transduction, magenta). GFP$^+$ and TH$^-$ (erroneous transduction, green) was counted as unspecific transduction. (**d**) The average specificity of GFP expression in the VTA of *Th^cre^* mice amounted to $61.7 \pm 6.8\%$ (derived from a total of 2,469 TH$^+$ and 1,627 eGFP$^+$ neurons obtained from 25 images of $n = 4$ mice). These findings are in line with the specificity of $59 \pm 1\%$ (dashed line) reported by Lammel and colleagues [39] (based on visual inspection). Numerical data underlying panel (d) can be found in S1 Data.
(TIF)

**S7 Fig. Comparison of native eGFP expression with anti-GFP staining.** (**a**) Example images of brain slices showing native eGFP fluorescence (green, top) and anti-GFP immunofluorescence with a secondary antibody in the red spectrum (magenta, center) in *Dbh^cre^*, *Net^cre^*, *Th^cre^*, and PRS×8 mice (from left to right). (**b**) Number of neurons detected by *CellPose* in the native eGFP (green, x-axis) and GFP-stained (red, y-axis) channels across animals ($n = 3/4/4/4$ *Dbh^cre^/Net^cre^/Th^cre^/*PRS×8 mice, respectively). $21.5 \pm 15.3\%$ more cells were detected in the GFP-stained channel as compared to the native eGFP channel (absolute difference: $20.5 \pm 16.5$ cells; $t$ test against 0: $t_{14} = 4.8$, $p = 2.8 \times 10^{-4}$). (**c**) Distribution of normalized fluorescence in anti-GFP stained channel as a function of the normalized fluorescence in the native eGFP channel across cells. Fluorescence was averaged within each cell mask (detected in the GFP-stained channel), then binned into $50 \times 50$ equally spaced bins, and further smoothed by a 2D $10 \times 10$ boxcar function. The averaged normalized brightness of immuno-stained cells per animal ($0.40 \pm 0.06$) exceeded the averaged normalized brightness of the native eGFP-expressing cells ($0.26 \pm 0.06$) by $55 \pm 35\%$ (absolute difference: $0.13 \pm 0.076$; $t$ test against 0: $t_{14} = 6.66$, $p = 1.08 \times 10^{-5}$).
(**d**) To quantify the difference in fluorescence between the anti-GFP immunostaining and native eGFP fluorescence in more detail, we fitted a non-linear function describing the fluorescence of GFP-stained cells as a function of native eGFP fluorescence (after within-channel normalization, see panel c). The function is of the form $f_R = \phi(\text{bias} + \text{gain} \times \phi^{-1}(f_G))$, where $f_R$ and $f_G$ denotes the fluorescence in the red (GFP-stained) and green (native eGFP) channels, respectively, and $\phi$ corresponds to the normal cumulative distribution function. The 'bias' parameter quantifies the overall change in fluorescence induced by the immunostaining (top panel), while the 'gain' parameter accounts for a possible asymmetry in the fluorescence increase depending on the native eGFP fluorescence (bottom panel). (**e**) Estimated parameter values from the function presented in (**d**). The 'bias' parameter was significantly larger than 0, indicating an increase in fluorescence following GFP-staining ($0.14 \pm 0.19$; $t$ test against 0: $t_{14} = 2.85$, $p = 0.013$). The 'gain' parameter was significantly smaller than 1, indicating that the increase in fluorescence was substantially stronger for the dimmest cells in the native eGFP channel ($0.50 \pm 0.17$; $t$ test against 1: $t_{12} = -11.56$, $p = 1.5 \times 10^{-8}$). In (**b**), (**c**), and (**f**), error bars denote the standard error of the mean (with $n = 15$), while statistical significance is denoted by */**/*** for $p < 0.05/0.01/0.001$, respectively. Numerical data underlying panel e can be found in S1 Data.
(TIF)

**S8 Fig. Ectopic transgene expression across model systems.** Example brain slices showing ectopic expression upon *Dbh^cre^* (**a**), *Net^cre^* (**b**), *Th^cre^* (**c–e**), and PRS×8 (**f**) mediated transgene expression. Immuno-straining against TH and GFP is displayed in magenta and green respectively. Insets show inverted gray scale images for a better contrast.
(TIF)

**S9 Fig. Effects of serotype and injection volume on viral spread.** Coronal brain slices of three wild-type mice that were bilaterally injected with 300 nl of rAAV2/2-hSyn-eGFP (top row), 300 nl of rAAV2/9-hSyn-eGFP (second row from top), 100 nl of rAAV2/9-hSyn-eGFP (third row from top), or 50 nl of rAAV2/9-hSyn-eGFP (bottom row). All slices were stained against tyrosine hydroxylase to visualize LC-NE neurons (magenta) and against eGFP to enhance the native fluorescence of transgene expression (green).
(TIF)

**S10 Fig. Supplementary readouts obtained in the open field test.** No effect of genotype could be detected on anxiety-like behavior as approximated by (**a**) the number of center crossings, (**b**) the latency to the first center crossing, (**c**) the mean minimum distance to the wall, (**d**) the percentage of time spent in the border of the arena, nor to general locomotion behavior as approximated by (**e**) the total distance traveled by the mouse. All data is depicted as mean ± standard error of the mean. n.s. = not significant. $n = 8$ females (circles) and 8 males (triangles). Numerical data underlying this figure can be found in S1 Data.
(TIF)

**S11 Fig. Supplementary readouts obtained in the elevated plus maze.** No effect of genotype could be detected on anxiety-like behavior as approximated by (**a**) the number of open arm entries, (**b**) the percentage of open arm entries on which the animal went to the distant edge of the arm, (**c**) the latency to leave the central platform, (**d**) the number of rearing events, (**e**) the latency to the first rearing event, or (**f**) the percentage of time spent grooming. All data is depicted as mean ± standard error of the mean. n.s. = not significant. $n = 8$ females (circles) and 8 males (triangles). Numerical data underlying this figure can be found in S1 Data.
(TIF)

**S12 Fig. Supplementary readouts obtained in the Y-maze.** No effect of genotype could be detected on locomotion or explorative behavior as approximated by (**a**) the average time per transition between Y-maze arms. All data is depicted as mean ± standard error of the mean. n.s. = not significant. $n = 8$ females (circles) and 8 males (triangles). Numerical data underlying this figure can be found in S1 Data.
(TIF)

**S13 Fig. Morris water maze during training sessions.** In each group ($n = 16$ mice/group), both the latency to reach the platform (**a**) and the distance until the platform was reached (**b**) decreased as a function of training session number (four sessions were performed on each day for two subsequent days), as indicated by both Pearson's and Spearman's correlation, indicating successful spatial learning in the water maze. Boxplots depict 0/25/50/75/100 percentiles, respectively. Dashed lines indicate the linear fit according to the method of least squares. Numerical data underlying this figure can be found in S1 Data.
(TIF)

**S14 Fig. Supplementary readouts obtained in the Morris water maze.** No effect of genotype could be detected on spatial memory as approximated by (**a**) the time spent in the platform area, (**b**) the time needed to reach the platform, and (**c**) the mean minimal distance to the platform. No effect of genotype could be detected on locomotion behavior as approximated by (**d**) the total distance traveled. No effect of genotype could be detected on (**e**) the average distance to the wall, suggesting comparable thigmotactic behavior of mice. All data is depicted as mean ± standard error of the mean. n.s. = not

significant. $n = 8$ females (circles) and 8 males (triangles). */**/***, depicted in panel a, for $p < 0.05/0.01/0.001$, respectively, tested against chance level. Numerical data underlying this figure can be found in S1 Data.
(TIF)

## Acknowledgments

The authors gratefully acknowledge Dr. Ingke Braren for production of viral constructs and Stefan Schillemeit for expert technical support with histology. We thank Prof. Dr. Bernardo Sabatini for kindly gifting the rAAV2/9-EF1a-DO-DIO-tdTomato-eGFP construct (*AddGene* plasmid #37120), the GENIE Project for gifting pGP-AAV-syn-jGCaMP8m-WPRE (*AddGene* plasmid #162375), Douglas Kim, *PhD*, and the GENIE Project for gifting pGP-CMV-NES-jRGECO1a (*AddGene* plasmid #61563), and Prof. Dr. Susumu Tonegawa for kindly providing the *Net^cre* line. We thank Patricia Jensen, *PhD*, and Nick Plummer, *PhD*, for helpful discussions on the different approaches of creating transgenic mouse lines. We thankfully acknowledge the support of the Core Facilities "Live Cell Imaging Mannheim (LIMa)" and "Preclinical Models" of the Medical Faculty Mannheim at Heidelberg University, and the data storage service SDS@hd supported by the Ministry of Science, Research and the Arts Baden-Württemberg (MWK) and the German Research Foundation (DFG) through grant INST 35/1503-1 FUGG.

## Author contributions

**Conceptualization:** Alexander Dieter, J. Simon Wiegert.

**Data curation:** Chantal Wissing, Lena S. Eschholz, Maxime Maheu, Alexander Dieter.

**Formal analysis:** Chantal Wissing, Lena S. Eschholz, Maxime Maheu, Alexander Dieter.

**Funding acquisition:** Fabio Morellini, J. Simon Wiegert.

**Investigation:** Chantal Wissing, Lena S. Eschholz, Kathrin Sauter, Alexander Dieter.

**Methodology:** Fabio Morellini, J. Simon Wiegert.

**Project administration:** J. Simon Wiegert, Alexander Dieter.

**Resources:** Fabio Morellini, J. Simon Wiegert.

**Software:** Maxime Maheu, Alexander Dieter.

**Supervision:** Fabio Morellini, J. Simon Wiegert, Alexander Dieter.

**Validation:** Chantal Wissing, Lena S. Eschholz, Maxime Maheu, Kathrin Sauter, Fabio Morellini, J. Simon Wiegert, Alexander Dieter.

**Visualization:** Lena S. Eschholz, Maxime Maheu, Alexander Dieter.

**Writing – original draft:** Alexander Dieter.

**Writing – review & editing:** Chantal Wissing, Lena S. Eschholz, Maxime Maheu, Kathrin Sauter, Fabio Morellini, J. Simon Wiegert, Alexander Dieter.

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
