## [Editor Report · Decision Letter 0]

12 Aug 2024

Dear Dr Dieter, 

Thank you for submitting your manuscript entitled "Targeting Norepinephrine Neurons of the Locus Coeruleus: A Comparison of Model Systems and Strategies" for consideration as a Short Reports by PLOS Biology.

Your manuscript has now been evaluated by the PLOS Biology editorial staff as well as by an academic editor with relevant expertise and I am writing to let you know that we would like to send your submission out for external peer review as a Methods and Resources paper. 

Once your full submission is complete, your paper will undergo a series of checks in preparation for peer review. After your manuscript has passed the checks it will be sent out for review. To provide the metadata for your submission, please Login to Editorial Manager (https://www.editorialmanager.com/pbiology) within two working days, i.e. by Aug 14 2024 11:59PM.

Kind regards,

Suzanne de Bruijn, PhD

Associate Editor, PLOS Biology

Sbruijn@plos.org

On behalf of,

Christian

Christian Schnell, PhD, 

Senior Editor

PLOS Biology

cschnell@plos.org

---

## [Decision Letter · Decision Letter 1]

6 Sep 2024

Dear Alexander,

Thank you for your patience while your manuscript "Targeting Norepinephrine Neurons of the Locus Coeruleus: A Comparison of Model Systems and Strategies" was peer-reviewed at PLOS Biology. It has now been evaluated by the PLOS Biology editors, an Academic Editor with relevant expertise, and by several independent reviewers. 

In light of the reviews, which you will find at the end of this email, we would like to invite you to revise the work to thoroughly address the reviewers' reports.

As you will see below, the reviewers have very positive comments about your manuscripts. Most of their concerns can be addressed through textual revision, although some of their points may require additional experimental data.

Given the extent of revision needed, we cannot make a decision about publication until we have seen the revised manuscript and your response to the reviewers' comments. Your revised manuscript is likely to be sent for further evaluation by all or a subset of the reviewers.

**IMPORTANT - SUBMITTING YOUR REVISION**

*Re-submission Checklist*

*Published Peer Review*

*PLOS Data Policy*

*Blot and Gel Data Policy*

Sincerely,

Christian

Christian Schnell, PhD

Senior Editor

PLOS Biology

cschnell@plos.org

REVIEWS:

Reviewer #1: In this paper, the authors conducted a comparison of different methods to genetically target LC NE neurons. They compared three genetically modified mouse lines: DBH-Cre, NET-Cre, and TH-Cre, as well as cell targeting using a synthetic PRS×8 promoter. Firstly, they compared the transduction efficiency (comprising efficacy and specificity) of the different approaches and found that DBH-Cre, NET-Cre, and PRS×8 induced more efficient transduction than TH-Cre. More interestingly, in terms of ectopic expression, DBH-Cre showed the lowest level compared to all the other models, with TH-Cre showing substantial ectopic expression in the peri-LC areas such as in the parabrachial nucleus. In addition, the authors screened the different Cre mouse lines behaviorally to investigate differences in stress and anxiety (using open field and EPM assays) and in the formation of episodic and spatial memory (using Y-maze and Morris water maze assays). No genotype-related behavioral differences were found in any of the behavioral assays considered. Overall, the data are very well presented. The authors conducted essential controls to verify their findings, and they effectively quantified the success rates of LC targeting. I believe that this study would make a great contribution to the LC field, and some of their findings might explain reproducibility issues among different laboratories studying the LC using different targeting approaches. Thus, I am in favor of publication. However, I have some comments which I think could further strengthen the results.

1) The substantial ectopic expression, especially in the peri-LC area, in the case of the TH-Cre mouse line, is very worrisome for its use in studying the LC, as the authors clearly explain in their discussion. It is interesting, however, that areas such as the parabrachial nucleus appear as sites of ectopic expression in both the TH-Cre and PRSx8-mediated expression cases. Do you think this is just by chance? Moreover, as the authors cite in their discussion, areas around the LC have been shown to express TH (although these might be dopaminergic rather than noradrenergic), which could explain the ectopic expression in the TH-Cre mouse line. However, those areas do not show up in TH staining. Do you think that the GFP expression in those peri-LC areas is induced only by leaky expression of Cre due to the way the TH-Cre mouse line was generated? Could one imagine that the TH-staining procedure has some limited sensitivity to those cells, and they are indeed expressing TH endogenously? In the native-GFP expression (Supplementary Figure 4), it seems that the presumably TH-negative cells medial to the right LC express eGFP more strongly than the LC cells. How do you explain this? In addition, the number of eGFP-positive cells in the LC (Figure 1) in the case of the TH-Cre mouse line is surprisingly low. Why do you think this is the case, given that those cells are TH-positive? How do you explain this? I think the authors address some of these concerns in their discussion, but given the strong message this conveys to the field—regarding one of the most used mouse lines for LC targeting—I think there must be no doubt about that specific result.

2) In the case of the jRGECOa fiber photometry and the issue of isosbestic excitation which the authors explained, another way to compute the F0 would be to fit a 5th degree polynomial to the red signal and use this as an F0. I do not think this would change their observations; it is just a proposition. 

Reviewer #2: Here, Wissing et al. present a comprehensive characterization of commonly used transgenic+viral strategies for targeting norepinephrine-expressing neurons in the rodent Locus Coeruleus (LC). Their justification for this work is two-fold: 1) LC is a brain region with diverse functionality and disease relevance but has been understudied due to the challenge of accurately targeting it in vivo. With advances in transgenic and recombinase-dependent AAVs, there is an increasing motivation to study LC function, and number of tools have been developed for researchers to use, but distinctions in their performance have not been succinctly performed. 2) Recent characterization of other neuromodulatory transgenic lines have revealed behavioral abnormalities caused by Cre expression, an important caveat that could compromise functional characterization of targeted cell types. To address these issues, the authors perform a series of experiments that include quantification of Cre-dependent AAV expression in LC across transgenic lines, behavioral assessment of transgenic lines, and introduction and functional testing of PRSx8-dependent calcium reporters developed in their lab. 

Overall, I thought this study was well done, with rigorous experiments and data analysis, careful consideration and discussion of various results and caveats, and well-organized text and figures. It was very enjoyable to read! A few specific strengths:

-I found the Cellpose-based analysis of LC very impressive! Due to the density of NE+ neurons in LC, it can be very challenging to accurately quantify cell numbers in this region, and thus, it is often not thoroughly done in published papers. I think the strategy presented here could be broadly useful to the field, especially as the authors also present validation/comparison with manually-analyzed images. I would encourage the authors to ensure that the methods are detailed enough such that others can replicate these pipelines for their own studies.

-The description of success rates regarding LC targeting was also thoroughly done. Again, I would suspect that this phenomenon has been observed by many in the field, but without such careful dissection of possible causes or suggested resolutions. 

-The authors summarize a wide range of studies succinctly, such that I anticipate this manuscript could serve as a useful resource for the field. For instance, their discussion of efficacies and titers of AAVs used across studies is very helpful.

However, I would also suggest a few areas where the manuscript could be strengthened to improve the rigor and usefulness of these studies. 

1) I think consideration of how serotypes and promoters affect LC infectivity needs to be addressed experimentally, rather than just discussed. I understand why it's not possible for the authors to extensively test all conditions that might influence efficacy and efficiency of LC infection, but even an initial comparison would greatly increase the usefulness of these experiments for the broader community: 

a) Experimental comparison of the results they report here with AAV2/9 with another commonly used serotype …maybe AAV2/2 since it has been reported to have more restricted spread (PMID: 25240284)?

b) Related, I think inclusion of an additional AAV reporter that utilizes a different promoter would greatly improve the study. I would suggest hSyn as many neural circuit-related AAV tools make use of it due to its small size. Also, I think it would be sufficient to compare performance of an hSyn-DIO-eGFP AAV to CAG-DIO-eGFP AAV in the AAV2/9 serotype only; no need to also expand this condition to multiple serotypes. 

2) I was confused by the PRSx8 results shown in Figure 1. In the image in 1c (bottom right corner), it looks like there's a lot of GFP+/TH- cells (so a lot of non-specific labeling), but this doesn't seem to be reflected in the metric reported in Figure 1d, where the samples seem to cluster much more closely to NETCre and DbhCre samples. I'm wondering if the authors expanded Figure 1 to include bar graphs of the specificity and efficacy values separately (or included this information as a supplementary figure) it would be easier to visualize specificity differences across samples and conditions? 

3) (minor). Some of these tools (ThCre and PRSx8) are also frequently utilized in rats, but no direct comparisons are made in the paper. I understand why it might be beyond the scope of the paper for the authors to experimentally test this (e.g., compare PRSx8-eGFP expression in mice vs. rats or DIO-eGFP expression in ThCre mice vs ThCre rats). But at a minimum, I think there could be some discussion on whether they expect the tools to perform differently across model systems based on published studies, especially since they do such a good job compiling data across papers in other topics of the discussion.

Reviewer #3 (Jorge Miranda-Barrientos): In the manuscript "Targeting Norepinephrine Neurons of the Locus Coeruleus: A Comparison of Model Systems and Strategies," Wissing and collaborators compare various methods for selectively targeting Locus Coeruleus Norepinephrine (LC-NE) neurons. The authors examine three of the most commonly used Cre-driver lines (DBH-Cre, TH-Cre, and NET-Cre), along with a recently developed synthetic viral construct that utilizes a promoter derived from the human DBH promoter (PRSx8). To assess selectivity and specificity in the Cre-driver lines and wild-type animals injected with the PRSx8 viral vector, the authors injected a Cre-dependent virus encoding eGFP into the LC and asses its expression into the LC. The authors report variations in efficacy and specificity across the different targeting methods, noting similar performance among the DBH-Cre, and PRSx8-virus, while observing markedly lower efficiency and specificity in Cre-specific viral transduction within TH-Cre animals. Furthermore, the authors observed no behavioral abnormalities (anxiety, fear, and working memory) in any of the Cre driver lines. Finally, the authors developed and validated two viral vectors with the PRSx8 promoter to track LC NE neuronal activity using GCaMP8m (Green spectrum) and GRECO (red spectrum). The data presented by Wissing and collaborators shows that selective targeting of LC-NE neurons is achievable using DBH-Cre, and PRSx8 viral transduction, while cautioning experimenters to consider viral vector serotypes, tropisms, and the external validity of these data relative to specific experimental designs.

The information presented in the manuscript is highly relevant, as selectively targeting LC-NE neurons is crucial for understanding the cellular mechanisms underlying a wide variety of behaviors involving LC-NE. Furthermore, the lack of specificity in tools aimed at selectively targeting neuronal populations can introduce significant confounding factors and therefore such tools need to be validated and direct contrasted between the different options available. Overall, the manuscript is well written, and the data presented is convincing. The manuscript is easy to follow, with most of the experiments being logical and well-controlled. The supplementary data presented demonstrate the rigorous controls for most of the experiments. That being said, there are a few considerations that the authors might want to address before publication.

Considerations: 

* The development of novel two viral vectors for assessing LC-NE neuronal activity using the PRSX8 platform is of great utility for the field. Although the pupillometry data gives support to the photometry data, it would be nice that the authors compare the data obtained with these novel viral vectors with a cre-dependent viral vector in one of the cre-driver lines that show similar specificity and efficacy to the PRSx8 viral vectors.

* In that same line, since the efficacy and specificity of the viral vectors depend on the construct and the proteins encoded, it would be nice if the authors show data about the specificity and efficacy of their developed viral vectors. 

* It would be informative if the authors included pictures showing the targeting of the optic fiber over the LC for the photometry recordings in figure 4.

* To aid clarity of results and data presentation, in Figure 1d the authors might want to consider switching the axis, so efficacy appears in the left X axis and specificity appears in the lower Y axis. This might help the readers to interpret the data presented as normally higher values are presented on the upper right-hand side of the graphs.

---

## [Editor Report · Decision Letter 2]

6 May 2025

Dear Alexander,

Thank you for your patience while we considered your revised manuscript "Targeting Norepinephrine Neurons of the Locus Coeruleus: A Comparison of Model Systems and Strategies" for publication as a Methods and Resources at PLOS Biology. This revised version of your manuscript has been evaluated by the PLOS Biology editors and the Academic Editor.

Based on our Academic Editor's assessment of your revision, we are likely to accept this manuscript for publication, provided you satisfactorily address the following data and other policy-related requests:

* We would like to suggest a different title to improve its accessibility for our broad audience: 

Comparison of approaches to label and manipulate norepinephrine neurons in the locus coeruleus reveals high variability between different systems

* Please include the approval/license number of the ethical approval for the animal experiments.

* DATA POLICY:

Regardless of the method selected, please ensure that you provide the individual numerical values that underlie the summary data displayed in the following figure panels as they are essential for readers to assess your analysis and to reproduce it: 1DEF, 3CD, 5ABCD, S2, S6D, S7E, S10ABCDE, S11ABCDEF, S12A, S13AB and S14ABCDE.

* CODE POLICY

We expect to receive your revised manuscript within two weeks. 

*Published Peer Review History*

*Press*

Sincerely,

Christian

Christian Schnell, PhD

Senior Editor

cschnell@plos.org

PLOS Biology

---

## [Editor Report · Decision Letter 3]

26 May 2025

Dear Alexander,

Thank you for the submission of your revised Methods and Resources "A comparison of viral strategies and model systems to target norepinephrine neurons in the locus coeruleus reveals high variability in transgene expression patterns" for publication in PLOS Biology. On behalf of my colleagues and the Academic Editor, Eric Nestler, I am pleased to say that we can in principle accept your manuscript for publication, provided you address any remaining formatting and reporting issues. These will be detailed in an email you should receive within 2-3 business days from our colleagues in the journal operations team; no action is required from you until then. Please note that we will not be able to formally accept your manuscript and schedule it for publication until you have completed any requested changes.

PRESS

Sincerely, 

Christian

Christian Schnell, PhD

Senior Editor

PLOS Biology

cschnell@plos.org